# Cytokinin Regulates Energy Utilization in *Botrytis cinerea*

Gautam Anand,[a] Rupali Gupta,[a] Maya Bar[a]

aDepartment of Plant Pathology and Weed Research, ARO, Volcani Institute, Rishon LeZion, Israel

**ABSTRACT** The plant hormone cytokinin (CK) is an important developmental regulator. Previous work has demonstrated that CKs mediate plant immunity and disease resistance. Some phytopathogens have been reported to secrete CKs and may manipulate CK signaling to improve pathogenesis. In recent work, we demonstrated that CK directly inhibits the development and virulence of fungal phytopathogens by attenuating the cell cycle and reducing cytoskeleton organization. Here, focusing on *Botrytis cinerea*, we report that CK possesses a dual role in fungal biology, with role prioritization being based on sugar availability. In a sugar-rich environment, CK strongly inhibited *B. cinerea* growth and deregulated cytoskeleton organization. This effect diminished as sugar availability decreased. In its second role, we show using biochemical assays and transgenic redox-sensitive fungal lines that CK can promote glycolysis and energy consumption in *B. cinerea*, both *in vitro* and *in planta*. Glycolysis and increased oxidation mediated by CK were stronger in low sugar availability, indicating that sugar availability could indeed be one possible element determining the role of CK in the fungus. Transcriptomic data further support our findings, demonstrating significant upregulation to glycolysis, oxidative phosphorylation, and sucrose metabolism upon CK treatment. Thus, the effect of CK in fungal biology likely depends on energy status. In addition to the plant producing CK during its interaction with the pathogen for defense priming and pathogen inhibition, the pathogen may take advantage of this increased CK to boost its metabolism and energy production, in preparation for the necrotrophic phase of the infection.

**IMPORTANCE** The hormone cytokinin (CK) is a plant developmental regulator. Previous research has highlighted the involvement of CK in plant defense. Here, we report that CK has a dual role in plant-fungus interactions, inhibiting fungal growth while positively regulating *B. cinerea* energy utilization, causing an increase in glucose utilization and energy consumption. The effect of CK on *B. cinerea* was dependent on sugar availability, with CK primarily causing increases in glycolysis when sugar availability was low, and growth inhibition in a high-sugar environment. We propose that CK acts as a signal to the fungus that plant tissue is present, causing it to activate energy metabolism pathways to take advantage of the available food source, while at the same time, CK is employed by the plant to inhibit the attacking pathogen.

**KEYWORDS** *Botrytis cinerea*, cytokinin, pathogenesis, nutrition, sugar, glycolysis, redox

Plant cytokinins (CKs) are known to be important in many aspects of plant life, including development of vasculature, differentiation of embryonic cells, seed development, maintenance of meristematic cells, growth and branching of roots, shoot formation, chloroplast biogenesis, and leaf senescence (1, 2). CKs also play an important role in nutrient balance and stress responses in the plant (3, 4). They are known to influence macronutrient balance by regulating the expression of nitrate, phosphate, and sulfate transporters (5–8). In addition, roles for CKs in fungal pathogenesis have also been suggested, either in the context of the pathogen producing CKs, or in the context of the pathogen activating the CK pathway in the host plant (9–12). Jameson (13) suggested that to achieve pathogenesis in the host, CK-secreting fungal biotrophs

Address correspondence to Maya Bar, mayabar@volcani.agri.gov.il.

The authors declare no conflict of interest.

or hemibiotrophs alter CK signaling to regulate the host cell cycle and nutrient allocation. For instance, germinating uredospores of *Puccinia* spp. have been shown to accumulate CK, modifying CK signaling to maintain the plant cell cycle (14, 15). Plant CK levels can be modulated by the application of exogenous CKs, and several studies have found a positive effect of CK treatment in the reduction of diseases caused by smut fungi, powdery mildew, and viruses (10, 16–18). More recently, CK was found to enhance disease resistance to non-obligatory plant pathogens. Elevated levels of CKs were shown to increase host resistance to various pathogens in a wide range of plants (4, 12, 19–21). We recently reported that endogenous increases in CK and exogenous applications of CKs induce systemic immunity in tomato, enhancing resistance against the fungal pathogen *Botrytis cinerea* through salicylic acid and ethylene signaling (4).

*B. cinerea*, the causative agent of gray mold disease, is a cosmopolitan pathogen that can infect more than 1,400 host plants and that causes massive losses worldwide annually (22). *B. cinerea* is a mostly (23, 24) necrotrophic pathogen, that has been widely used as a model pathogen to study various mechanism underlying plant-pathogen interactions. Due to its economic importance, *B. cinerea* is listed in the top 10 important plant fungal pathogens (25). During pathogenesis, *B. cinerea* induces necrosis in the host by producing various toxins such as botrydial, botcinic acid, and its derivatives (26, 27) and reactive oxygen species (ROS). *B. cinerea* also manipulates the host plant into generating oxidative bursts that facilitate colonization (28, 29), and promote extension of macerated lesions by the induction of apoptotic cell death. Various enzymes, including lytic enzymes, which are sequentially secreted by the fungus, facilitate penetration and colonization and produce an important source of nutrients for the fungus (30).

*B. cinerea* spores are primary sources of infection in plants in nature. After contacting the plant surface, the spores germinate to form short germ tubes that directly penetrate plant tissues (31). It has long been known that germination of spores and infection through plant surfaces relies on spore ability to access the nutrient supply offered by living plants (32, 33). Solomon et al. (34) have suggested a model that describes nutrient availability to fungi during different phases of fungal infection of a plant host. The first phase, which involves spore germination and host penetration, is based on lipolysis. The second phase, which requires invasion of plant tissues, uses glycolysis. Studies on *Tapesia yellundae*, *Colletotrichum lagenarium*, and *Cladosporium fulvum* suggest that lipids are the main source of energy during germination and penetration, while after penetration, the available plant sugars become the main source of energy (35–37). These different stages of fungal infection and metabolism depend on nutrient availability and allocation. The role of CK in nutrient balance in the plant is well studied, but its possible role in nutrient balance and metabolism in necrotrophic fungi such as *B. cinerea* needs to be investigated.

Recently, we have found that CK inhibits the growth of a variety of plant-pathogenic fungi. In-depth characterization of the phenomenon in *B. cinerea* revealed that CK in plant physiological concentrations can inhibit the sporulation, spore germination, and virulence (38) of *B. cinerea*. We also found CK to affect both budding and fission yeast. Transcriptome profiling of *B. cinerea* grown with CK revealed that DNA replication and the cell cycle, cytoskeleton integrity, and endocytosis, are all repressed by CK (38).

Given that CK had such fundamental, conserved, and ubiquitous effects on fungal development, the question of a possible role for CK in affecting fungal metabolism and nutrition, in particular during host-pathogen interactions, arises. In this study, we investigated the effect of CK on fungal metabolism. Using *B. cinerea*, we examined how CK affects fungal metabolism and nutrition, both in the context of fungal growth and during infection in the tomato host. We found that CK promoted fungal metabolism, with inverse correlation to sugar availability, and that the redox sate of *B. cinerea* was affected by CK both during growth and during infection in the plant host. Our results reveal additional roles for CK in fungus-plant interactions and may shed light on the availability of energy and nutrients to the fungus during the initial stages of plant infection.

## RESULTS

**CK-mediated *B. cinerea* growth inhibition and cytoskeleton deregulation are affected by sugar and nutrient availability.** In order to examine whether the effect of CK on *B. cinerea* is related to nutrient or sugar availability, we grew *B. cinerea* in different synthetic media, with and without the CK 6-benzylaminopurine (6-BAP). The different media are detailed in Table 1 in the Materials and Methods section. The inhibition of mycelial growth by CK was found to depend on sugar availability. Dilution of glucose to one-quarter of the level included in rich medium resulted in a reversal of CK-mediated growth inhibition, causing growth promotion (Fig. 1). Dilution of minerals (nitrogen, phosphate) to one-quarter of the level included in rich medium resulted in some general growth inhibition but did not affect CK-mediated growth inhibition. Dilution of both sugar and minerals behaved similarly to dilution of sugar alone, resulting in CK-mediated growth promotion, suggesting that the sugar effect is dominant. Similar assays were performed in potato dextrose agar (PDA) and potato dextrose broth (PDB) media (Fig. S1 and S2), in which total medium dilution resulted in gradual decreases in CK-mediated growth inhibition, likely due to the combined restriction in both sugars and minerals, similar to medium 4 (1/4 Gluc, N, P) in Fig. 1. As the experiments presented in Fig. 1 were conducted in the dark, we conducted identical assays with fungi grown in light, with similar results (Fig. S3). Further, we also examined changes to medium pH in all our assays, to confirm that the observed growth effects are not a result of pH alterations (Fig. S4). CK promoted medium acidification in all cases. Although medium acidification was slightly stronger in media 3 and 4, the levels were similar in full medium (medium 1) and medium with restricted glucose (medium 2), suggesting that the medium pH is not the source of the observed alterations in CK activity upon glucose restriction. Growth inhibition of *B. cinerea* by CK in rich medium was previously reported by our group (38).

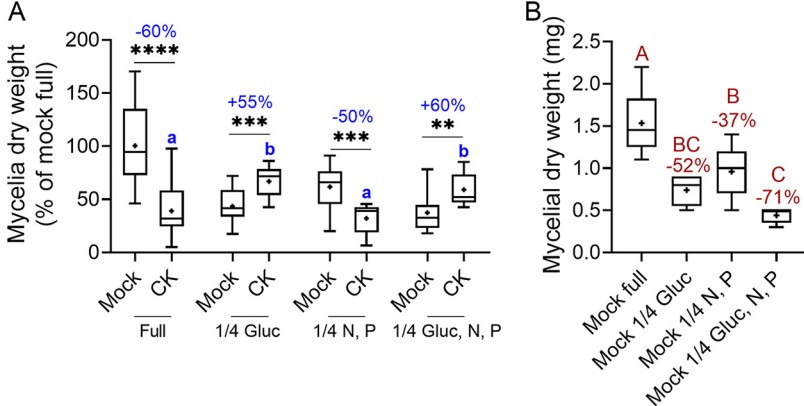

**FIG 1** Cytokinin (CK)-mediated growth effects depend on glucose availability. *B. cinerea* spores were inoculated in four types of defined media: full (20 g/liter glucose, and 4 g/liter each of $K_2HPO_4$, $KH_2PO_4$, and $(NH_4)_2SO_4$), 1/4 glucose (5 g/liter glucose and 4 g/liter each of $K_2HPO_4$, $KH_2PO_4$, and $(NH_4)_2SO_4$), 1/4 nitrogen and phosphate (20 g/liter glucose and 1 g/liter each of $K_2HPO_4$, $KH_2PO_4$, and $(NH_4)_2SO_4$), and 1/4 glucose, nitrogen, and phosphate (5 g/liter glucose and 1 g/liter each of $K_2HPO_4$, $KH_2PO_4$, and $(NH_4)_2SO_4$). The samples were prepared without (mock) or with the addition of the CK 6-benzylaminopurine (6-BAP, 100 $\mu$M). Germinated spores were grown with shaking (150 rpm) at 22 $\pm$ 2°C in the dark, for 3 days, after which the fungal matter was dried and weighed. Box plots are shown with minimum to maximum values, inner quartile ranges (box), median (line in box), mean (plus sign in box), and outer quartile ranges (whiskers). The results were analyzed for statistical significance using Welch's analysis of variance (ANOVA) with Dunnett's *post hoc* test or a two-tailed *t* test with Welch's correction. (A) Asterisks indicate statistically significant differences between the mock and CK samples within the same medium. ****, $P < 0.0001$; ***, $P < 0.001$; **, $P > 0.01$ ($N = 12$). Different lowercase letters indicate statistically significant differences in the growth of CK-treated samples in different media ($P < 0.02$). Percentages of the effect of CK on fungal growth in the different media are indicated above the asterisks. (B) Comparison of mock samples in the different media. Different uppercase letters indicate statistically significant differences between mock samples grown in the indicated medium. The percentage of growth inhibition in the different media compared with the full medium is indicated above the upper whisker for each medium ($N = 6$). $P < 0.027$.

To examine cytoskeleton integrity, we transformed *B. cinerea* with Lifeact-GFP (39) and proceeded to treat the transformed fungal cells with CK (6-BAP) in full and 1/4 PDB (Fig. 2), as the Lifeact-transformed *B. cinerea* has abnormal growth (39) and did not grow well in our defined media. As reported previously by our group (38), we observed mislocalization of actin, which is normally localized to growing hyphal tips (40, 41), upon CK treatment in full PDB. Fungal cells grown in 1/4 medium displayed reduced actin polarization compared with those grown in full medium (Fig. 2), resulting in a smaller, although still significant, effect of CK on tip-specific localization of F-actin in 1/4 medium (Fig. 2). Reduced actin polarization upon media dilution and CK treatment suggests that actin depolarization is one of the underlying mechanisms of nutrient-restriction-mediated growth reduction and CK-mediated growth reduction.

The observation that CK-mediated growth inhibition is affected by sugar and mineral availability (Fig. 1 and 2; Fig. S1 and S2) prompted us to examine whether this is a general phenomenon in *B. cinerea*. To do this, we examined the correlation between growth rate and CK-mediated growth inhibition in 17 agricultural *B. cinerea* isolates from 6 different plant hosts (Fig. S5). We found a significant positive linear correlation between growth rate and CK-mediated growth inhibition (Fig. S5B), although additional factors undoubtedly also influence the CK sensitivity of different *B. cinerea* isolates.

**Transcriptome profiling reveals an effect of CK on *B. cinerea* metabolic pathways and sugar transport.** We previously conducted transcriptome profiling on *B. cinerea* treated with CK (6-BAP), finding downregulation upon CK treatment of a variety of growth and developmental pathways in *B. cinerea*, including inhibition of cell division, DNA replication, endocytosis, and the actin cytoskeleton (38). Given that we found the effect of CK to depend on the nutritional context (Fig. 1), we next mined our transcriptomic data for alterations in gene expression that might explain this phenomenon. Interestingly, we found that the glycolysis, sucrose metabolism, and oxidative phosphorylation pathways, according to the Kyoto Encyclopedia of Genes and Genomes (KEGG), were significantly upregulated in *B. cinerea* upon CK treatment (38) (Fig. 3; Data Set S1). Over a quarter of the pathway genes were upregulated in the glycolysis (Fig. 3A) and oxidative phosphorylation (Fig. 3C) pathways, with an false discovery rate (FDR) corrected *P* value of <0.0071 for glycolysis and of <2.94$^{E-11}$ for oxidative phosphorylation. A third of the sucrose/starch metab-

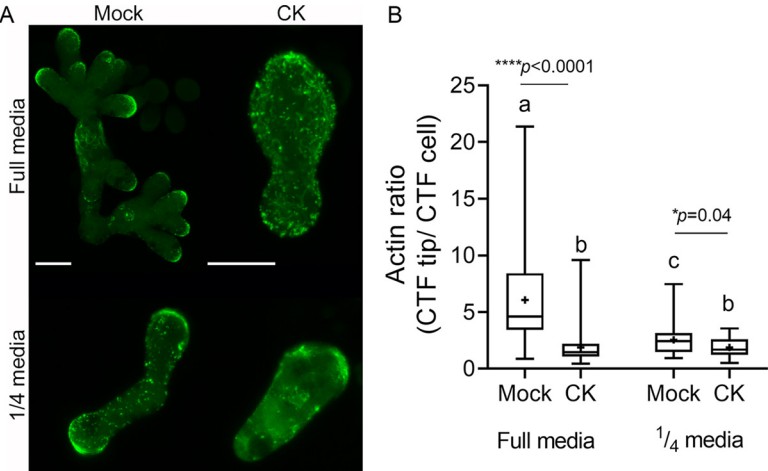

**FIG 2** CK and nutrient availability affect actin distribution. Spores of *B. cinerea* expressing the filamentous actin marker Lifeact-GFP were treated with mock or CK (6-benzylaminopurine, 100 $\mu$M) and grown for 6 and 24 h prior to confocal visualization, in full and one-fourth potato dextrose broth medium, respectively. (A) Representative images. Bar, 10 $\mu$M. (B) Analysis of corrected total fluorescence (CTF) of the ratio between actin at the tip of the cell and the total cell in mock- and CK-treated cells. Three independent experiments were conducted with a minimum of 24 images analyzed ($N > 32$ growing hypha tips). Box plots are shown with minimum to maximum values, inner quartile ranges (box), median (line in box), mean (plus sign in box), and outer quartile ranges (whiskers). Letters and asterisks indicate significance in Kruskal-Wallis ANOVA with Dunn's *post hoc* test. *, $P < 0.05$; ****, $P < 0.0001$.

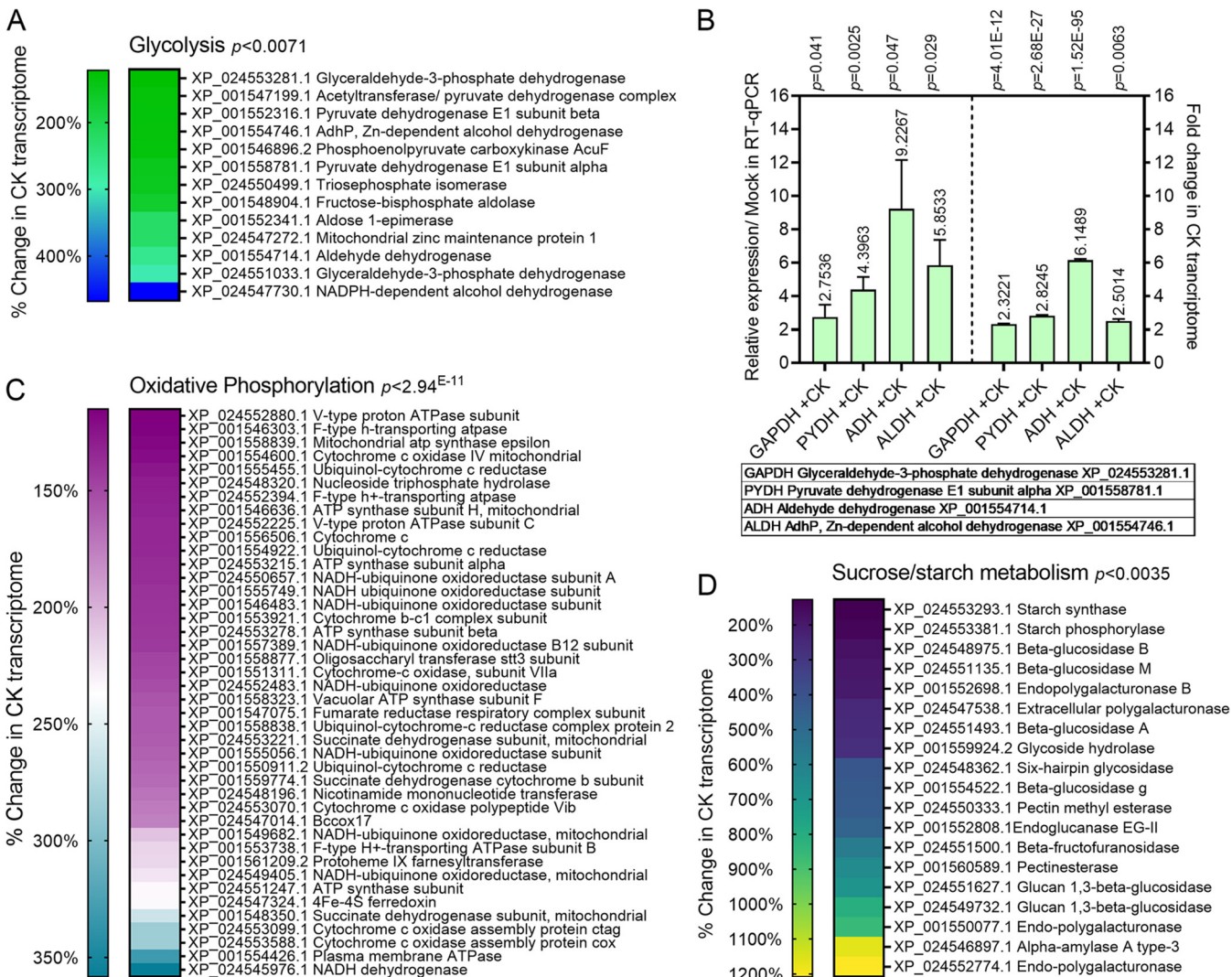

**FIG 3** Transcriptomic analysis of *Botrytis* grown with CK reveals upregulation of energy metabolism pathways. Illumina Hiseq next-generation sequencing (NGS) was conducted on *B. cinerea* mock-treated or CK-treated (6-benzylaminopurine, 25 $\mu$M) samples, with three biological repeats each. Gene expression values were computed as fragments per kilobase per million (FPKM), and differential expression analysis was completed using the DESeq2 R package. Genes with an adjusted *P* value (padj) of no more than 0.05 and log$_2$FC (where FC indicates fold change) greater than 1 or less than −1 were considered differentially expressed. The KOBAS 3.0 tool was used to detect the statistical enrichment of differential expression genes in Kyoto Encyclopedia of Genes and Genomes (KEGG) pathways and Gene Ontology (GO). The pathways were tested for significant enrichment using Fisher's exact test, with Benjamini and Hochberg false discovery rate (FDR) correction. The corrected *P* value was deemed significant at *P* < 0.05. The glycolysis (A, B), oxidative phosphorylation (C), and sucrose metabolism (D) pathways were all found to be significantly upregulated upon CK treatment (see also Supplemental Data 1). (A, C, D) Heat map representation of upregulated genes in the CK transcriptome in each indicated pathway. (B) Comparison of reverse transcription-quantitative PCR (RT-qPCR) validation of the four indicated key glycolysis genes with the transcriptomic values. Expression levels are indicated above the bars, and *P* values of the difference from mock samples are indicated above the box for each sample (RT-qPCR: *t* test, Welch's correction; transcriptomic data: padj). The full transcriptome data were previously published (38) and is available (NCBI bioproject PRJNA718329).

olism pathway was upregulated, with an FDR-corrected *P* value of <0.0035 (Fig. 3D; Data Set S1). Interestingly, although we found virulence genes as a group to be downregulated upon CK treatment (38), sugar-metabolism genes known to have a role in virulence such as pectin methyl esterase and polyendogalacturonase (42) were upregulated upon CK treatment, despite the overall downregulation of virulence (Fig. 3D; Data Set S1). Sugar transporters are known to be upregulated in *B. cinerea* during pathogenesis (43, 44). In addition to glycolysis, sucrose metabolism, and oxidative phosphorylation, we found a significant upregulation of sugar transporter expression following CK treatment (38) (Data Set S1).

For glycolysis pathway genes, we also conducted a reverse transcription-quantitative PCR (RT-qPCR) validation of the upregulation of some of the key pathway genes found to be upregulated in the transcriptome following CK treatment. Fig. 3B depicts a

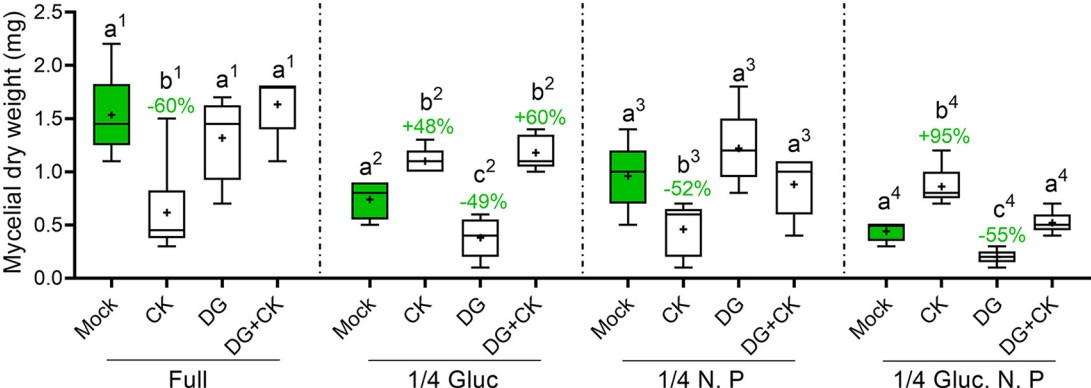

**FIG 4** CK rescues glycolysis inhibition under glucose restriction. *B. cinerea* spores were grown in four types of defined media: full (20 g/liter glucose and 4 g/liter each of $K_2HPO_4$, $KH_2PO_4$, and $(NH_4)_2SO_4$), 1/4 glucose (5 g/liter glucose and 4 g/liter each of $K_2HPO_4$, $KH_2PO_4$, and $(NH_4)_2SO_4$), 1/4 nitrogen and phosphate (20 g/liter glucose and 1 g/liter each of $K_2HPO_4$, $KH_2PO_4$, and $(NH_4)_2SO_4$), and 1/4 glucose, nitrogen, and phosphate (5 g/liter glucose and 1 g/liter each of $K_2HPO_4$, $KH_2PO_4$, and $(NH_4)_2SO_4$). The samples were prepared without (mock) or with the addition of the CK 6-benzylaminopurine (6-BAP, 100 $\mu$M), the competitive glucose inhibitor 2-deoxyglucose (2-DG, 2.5 mM), or both. Germinated spores were grown with shaking (150 rpm) at 22 $\pm$ 2°C in the dark for 3 days, after which the fungal matter was dried and weighed. Box plots are shown with minimum to maximum values, inner quartile ranges (box), median (line in box), mean (plus sign in box), and outer quartile ranges (whiskers) (N = 6). The results were analyzed for statistical significance using a two-tailed *t* test, with Welch's correction where appropriate. Different lowercase letters tagged with superior numbers indicate statistically significant differences between samples grown in the indicated medium, respectively. Where statistically significant differences from mock samples within a respective medium were observed, the percentage of growth promotion or growth inhibition is indicated above the upper whisker. $P < 0.045$.

comparison between the fold change of these genes in the transcriptome and the changes we observed independently in qPCR.

To further examine the effect of CK on *B. cinerea* metabolism, we used 2-deoxy-D-glucose (2-DG) and oligomycin (OM), which are inhibitors of glycolysis and ATP synthesis, respectively. 2-DG is a glucose analog that inhibits glycolysis by competing with glucose as a substrate for hexokinase, the rate-limiting enzyme in glycolysis. After entering the cell, 2-DG is phosphorylated by hexokinase II to 2-deoxy-D-glucose-6-phosphate (2-DG-6-P), but unlike glucose, 2-DG-6-P cannot be further metabolized by phosphoglucose isomerase. This leads to the accumulation of 2-DG-P in the cell and subsequent depletion in cellular ATP (45–47). *B. cinerea* was grown with 2-DG in the four different synthetic media detailed in Table 1. 2-DG significantly inhibited growth in media 2 and 4, in which glucose was diluted to one-quarter of its amount in rich medium (Fig. 4). CK (100 $\mu$M 6-BAP) was found to rescue the inhibitory effect of 2-DG when added to these diluted glucose media (Fig. 4). Lowering only the mineral availability, as in medium 3, did not result in DG-mediated growth inhibition or CK-mediated rescue. Growth rates of *B. cinerea* in the different media without CK are provided in Fig. 1. Similar results were obtained in PDA, where CK was able to rescue DG-mediated growth inhibition in diluted medium, but not in full medium (Fig. S6A).

OM inhibits mitochondrial $H^+$-ATP-synthase and has been attributed antifungal properties (46, 48). We were not able to examine OM activity in our defined media, because growth inhibition proved too strong. When examined in PDA at different medium dilutions, OM was inhibitory at all medium strengths, irrespective of the medium dilution. CK partially rescued OM-mediated growth inhibition at all medium strengths examined (Fig. S6B). These results support the notion that CK affects glycolysis in the fungus, possibly in the same pathway as ATP synthesis.

**CK induces glucose utilization in *B. cinerea*.** The dependence of CK-mediated growth inhibition on sugar availability, and the rescue of glycolysis inhibition by CK, indicated that CK is affecting *B. cinerea* metabolism. To further confirm this hypothesis, we next measured glucose utilization in the presence of CK (6-BAP) in both synthetic defined liquid media and PDB. We observed a significant increase of glucose utilization in the presence of CK in all media types, with the exception of the medium that

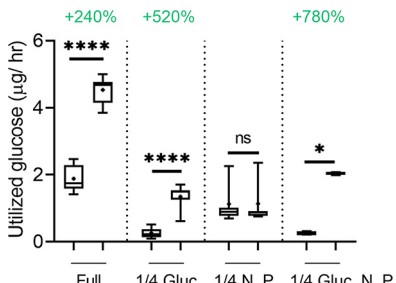

**FIG 5** CK increases glucose utilization. *B. cinerea* was grown from spores ($10^6$/mL) in four types of defined media: full (20 g/liter glucose and 4 g/liter each of $K_2HPO_4$, $KH_2PO_4$, and $(NH_4)_2SO_4$), 1/4 glucose (5 g/liter glucose and 4 g/liter each of $K_2HPO_4$, $KH_2PO_4$, and $(NH_4)_2SO_4$), 1/4 nitrogen and phosphate (20 g/liter glucose and 1 g/liter each of $K_2HPO_4$, $KH_2PO_4$, and $NH_4SO_4$), and 1/4 glucose, nitrogen, and phosphate (5 g/liter glucose and 1 g/liter each of $K_2HPO_4$, $KH_2PO_4$, and $(NH_4)_2SO_4$) with 150 rpm shaking at 22 ± 2°C in the dark, without (mock) or with the addition of the CK 6-benzylaminopurine (6-BAP, 100 $\mu$M). The amount of glucose in each medium was examined after 48 h and subtracted from the amount of glucose present in the medium without fungi that underwent similar treatment. The approximate percentage of increase in glucose utilization in the presence of CK is indicated above the bars in green for each medium. Box plots are shown with minimum to maximum values, inner quartile ranges (box), median (line in box), mean (plus sign), and outer quartile ranges (whiskers) ($N = 12$). The results were analyzed for statistical significance using a two-tailed *t* test with Welch's correction. Asterisks indicate a statistically significant increase in metabolized glucose upon CK treatment. ****, $P < 0.0001$; *, $P < 0.05$. ns, nonsignificant.

contained high sugar but diluted minerals (Fig. 5; Fig. S5). Interestingly, the percentage of increase of glucose utilization by *B. cinerea* in the presence of CK increased with decreasing sugar availability in the medium (Fig. 5; Fig. S5). 2-DG competes with glucose in the glycolytic pathway, suggesting that this increase in sugar utilization upon CK treatment might be the reason why CK rescues glycolysis inhibition by 2-DG (Fig. 4). The control adenine, which is structurally similar to 6-BAP, did not affect glucose utilization in *B. cinerea* (Fig. S7).

**CK alters *B. cinerea* redox status.** CK increased medium acidification during fungal growth in all tested synthetic media (Fig. S4). Since metabolic pathway fluctuations can affect redox status (47) and redox homeostasis in *B. cinerea* is known to change during host infection (49, 50), we next examined cytosolic and mitochondrial redox status in the fungus grown with CK (6-BAP). For this purpose, we generated *Bcl*-16 strain lines expressing GRX-roGFP and mito-roGFP, using previously described expression cassettes that were used for the measurement of *B. cinerea* redox status (39, 51). We found that after 24 h of growth *in vitro*, CK significantly altered the cytosolic redox state of *B. cinerea* to a more reduced state, while the mitochondrial redox state was significantly more oxidized with CK (Fig. 6A). Following the redox state over time, we observed similar states in the first 8 h of growth, with CK starting to affect the redox state after about 15 h of cocultivation, corresponding with the stage at which mycelia are elongating (Fig. 6B and C). Cytosolic and mitochondrial redox states are often inversely correlated (52). Interestingly, there was an inverse effect of CK on cytosolic and mitochondrial redox state of growing mycelia (Fig. 6).

**Endogenous CK content of tomato leaves affects redox state of *B. cinerea* during plant infection.** Since CK affected redox state in *B. cinerea* in rich medium, we next examined whether endogenous CK content in tomato leaves can affect the redox status of infecting *B. cinerea* mycelia. For this, *B. cinerea* GRX-roGFP and mito-roGFP conidia from freshly sporulated PDA plates were used to infect detached leaves from M82, isopentenyl transferase (IPT) overexpressing, and cytokinin oxidase (CKX) overexpressing plants. Leaves overexpressing IPT have increased CK content and are more resistant to *B. cinerea* infection, while leaves overexpressing CKX have reduced CK content and are more sensitive to *B. cinerea* infection (4). Redox-dependent changes in GRX-roGFP2 and mito-roGFP fluorescence in living *Botrytis* hyphae have been previously visualized by confocal laser scanning microscopy (CLSM) (52). Infecting *B. cinerea* hyphae expressing GRX-roGFP in the cytosol or mito-roGFP in the intermembrane mitochondrial space were analyzed microscopically, 24 and

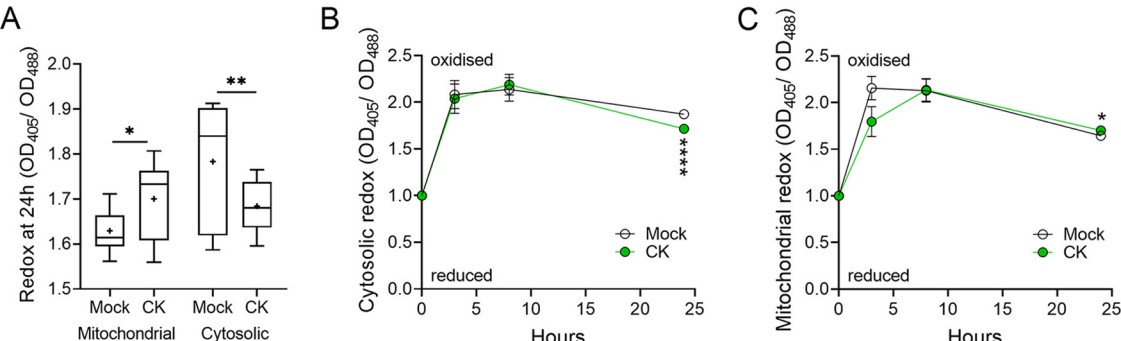

**FIG 6** CK alters *B. cinerea* redox state *in vitro* in rich medium. The redox state of *B. cinerea* without (mock) or in the presence of CK was assessed using roGFP transformed *B. cinerea*. Spores ($10^6$/mL) of *B. cinerea* strains expressing GRX-roGFP, for assessing cytosolic redox state, and mito-roGFP, for assessing mitochondrial redox state, were incubated in potato dextrose broth (PDB) without (mock) or with CK 6-benzylaminopurine (6-BAP, 100 $\mu$M), for 24 h at 18°C, with 150 rpm shaking. Fluorescence was measured using a fluorimeter, with excitation at 405 $\pm$ 5 nm for the oxidized state and 488 $\pm$ 5 nm for the reduced state of roGFP2. The emission was detected at 510 $\pm$ 5 nm. The redox ratio of the fungus was calculated as $Em_{405}/Em_{488}$ of relative fluorescence units (RFU). (A) Redox status of the mitochondria and cytosol, with and without CK, after 24 h. Box plots are shown with minimum to maximum values, inner quartile ranges (box), median (line in box), mean (plus sign in box), and outer quartile ranges (whiskers) ($N = 6$). (B) Time course of redox state in the cytosol. (C) Time course of redox state in the mitochondria. Asterisks indicate statistical significance in a two-tailed $t$ test. *, $P < 0.05$; **, $P < 0.01$; ****, $P < 0.0001$.

48 h after inoculation. Similar to the fluorometry-based calculations, a higher 405 nm/488 nm ratio indicates a more oxidized state, and a lower ratio indicates a more reduced state. We found that 24 h after inoculation, the cytosolic redox state of the infecting hyphae was more oxidized on IPT leaves (high CK content) compared to the infecting hyphae on mock M82 leaves, while infecting hyphae on CKX (low CK content) were more reduced (Fig. 7A and C). In a parallel set of experiments that included additional genotypes, we also observed increased oxidation of the *B. cinerea* cytosol when infecting M82 leaves that were pretreated with CK or when infecting leaves of the CK-hypersensitive *clausa* mutant (Fig. S8). At 48 h postinoculation, we observed an opposite trend of the cytosolic redox state of the infecting hyphae, with hyphae on IPT overexpressing leaves becoming more reduced, while hyphae on CKX overexpressing leaves were more oxidized, compared with M82 leaves. (Fig. 7A and C). Here, again, in a parallel set of experiments, which included additional genotypes, we also observed increased reduction of the *B. cinerea* cytosol when infecting leaves of the hypersensitive *clausa* mutant (Fig. S8). When examining the mitochondrial redox state of the hyphae growing on IPT overexpressing or CKX overexpressing leaves, we found that 48 h postinoculation, the mitochondrial redox state of the hyphae growing on IPT was significantly oxidized compared to mock M82, while hyphae growing on CKX were significantly reduced (Fig. 7B and C). The cytosolic redox state was measured in a parallel set of experiments on leaf discs cocultivated with *B. cinerea* spores in a fluorimeter, with similar results (Fig. S9). To verify that the virulence of the roGFP fungi was intact, we also conducted disease assays of these fungi on the different genotypes, with findings consistent with previous results (4): i.e., reduced disease on high-CK or CK-hypersensitive genotypes, and increased disease on low-CK genotypes (Fig. S10).

We further examined the transcriptome of *B. cinerea* grown in the presence of tobacco seedlings following CK treatment, finding significant changes, both significant downregulation and significant upregulation, in NADPH/NADH reductases and oxidoreductases (Fig. S11). These transcriptional changes in the fungus further support the notion that CK affects the ROS coping mechanisms in *B. cinerea*, and could underlie the altered pathogenesis courses observed in tomato genotypes with altered CK content.

**Altered response to CK in different media is not due to altered CK internalization.** To confirm that CK penetration into fungal cells is not reduced by diluting medium sugar or mineral concentration, we examined internalization of the CK isopentenyl (iP) tagged with the fluorophore NBD, which was previously used in CK internalization assays and demonstrated to retain CK activity (53). Due to the limited availability of the labeled CK, we conducted the assay in full and 1/4 PDB only. Fig. 8 demonstrates that the amount of iP-NBD found in the fungal cells after 2 h of incubation is similar in full and diluted

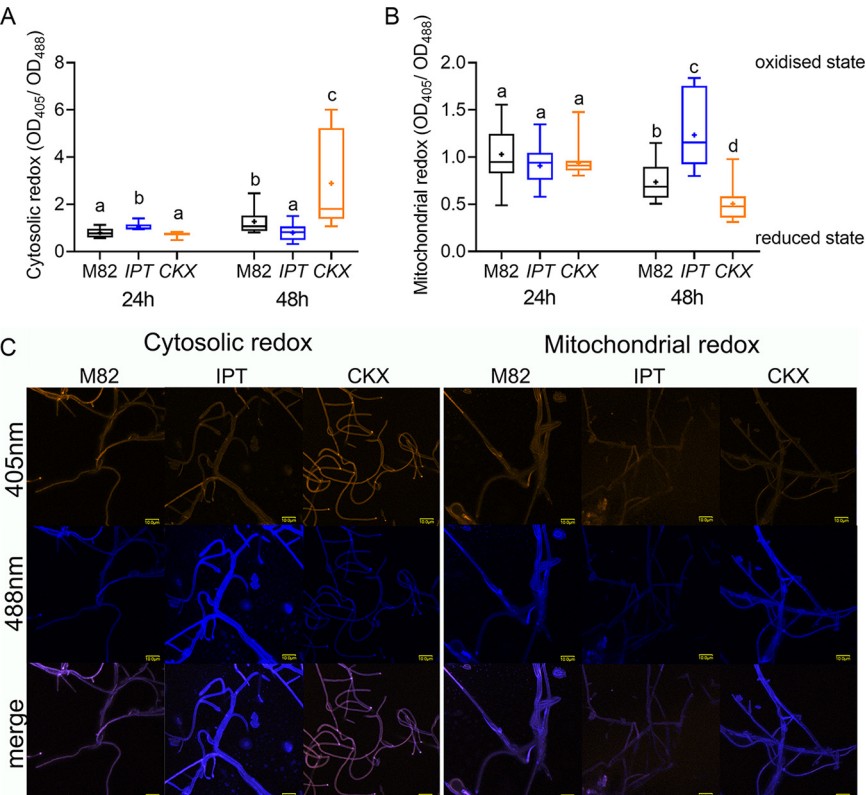

**FIG 7** Plant endogenous CK alters *B. cinerea* redox state during infection. The redox state of *B. cinerea* when infecting leaves of different CK-content tomato genotypes was assessed using roGFP transformed *B. cinerea*. Spores (10$^6$/mL in glucose and K$_2$HPO$_4$) of *B. cinerea* strains expressing GRX-roGFP for assessing the cytosolic redox state and mito-roGFP for assessing the mitochondrial redox state were used to infect the background M82 wild-type line, the high-CK *pBLS* ≫ *IPT7* overexpressing line (IPT), and the low-CK *pFIL* ≫ *CKX3* overexpressing line (CKX). *B. cinerea* fluorescence was captured using a confocal laser scanning microscope at 24 and 48 h, with excitation at 405 nm for the oxidized state and 488 nm for the reduced state of roGFP2. The emission was detected using a 505- to 530-nm bandpass filter. The redox ratio of the fungus was calculated as Em$_{405}$/Em$_{488}$ using ImageJ, from at least 12 images per time point, per treatment. (A) Redox status of the *B. cinerea* cytosol, after 24 and 48 h. (B) Redox status of the *B. cinerea* mitochondria, after 24 and 48 h. (A, B) Box plots are shown with minimum to maximum values, inner quartile ranges (box), median (line in box), mean (plus sign in box), and outer quartile ranges (whiskers) (*N* = 12). Differences between samples were assessed using one-way ANOVA with a Dunnett's *post hoc* test. Different letters indicate statistically significant differences between samples. (A) *P* < 0.021. (B) *P* < 0.029. (C) Representative images of the roGFP fungi growing on leaves of the different genotypes, captured at the "reduced" and "oxidized" wavelengths at 48 h. Bar, 10 μM.

media (Fig. 8A to D). CK can be internalized by endocytosis or via purine permeases (PUPs). We therefore also investigated whether CK treatment and/or medium dilution could affect the expression levels of PUPs in *B. cinerea*. With the exception of the combination of medium dilution and CK inducing the expression of PUP3, PUPs were essentially not significantly affected by medium dilution or CK treatment (Fig. 8E), suggesting that altered PUP expression probably does not affect CK internalization.

## DISCUSSION

It has been previously reported by us and others that CK promotes fungal disease resistance in plants (4, 12). Recently, we reported a direct inhibitory effect of CK on *B. cinerea* growth and development *in vitro* (38). Our previous results indicated that *B. cinerea* responds to CK and activates signaling cascades in its presence, leading to inhibition of the cell cycle, mislocalization of the actin cytoskeleton, and inhibition of cellular trafficking (38). We hypothesized that CK may exert its effect through influence on fungal metabolic pathways. The present study was performed to examine the effect of CK

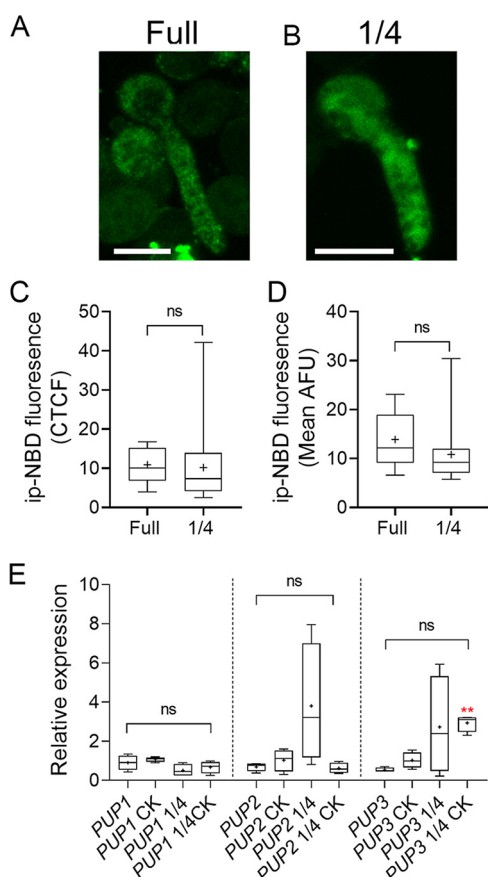

**FIG 8** Glucose-dependent alterations to *B. cinerea* CK response are not due to altered CK penetration. (A to D) *B. cinerea* was grown from spores ($10^6$/mL) in liquid PDB at full strength for 6 h or 1/4 strength for 24 h. 10 μM iP-NBD was then added to the germlings, which were grown with the fluorescent CK for an additional 2 h. The fungal matter was then washed in water and visualized under a confocal microscope. (A, B) Representative images. Bar, 10 μM. (C, D) Internalized CK was quantified from 17 images per treatment. (C, D) Differences in corrected total cell fluorescence (C, CTCF) and mean fluorescence as expressed in Arbitrary Fluorescent Units (D, AFU) were assessed using a Mann-Whitney U test. ns, not significant. (E) qRT-PCR comparison of purine permeases (PUPs) gene expression in fungi grown in full or 1/4 PDB, with our without CK (6-BAP, 100 μM). Differences in the expression of each PUP among the different samples were assessed using a Kruskal-Wallis ANOVA with Dunn's *post hoc* test and Welch's *t* test. No significant differences were observed in PUP gene expression, except for an increase in PUP3 in the 1/4 PDB + CK sample, compared to mock grown in full PDB ($N = 5$). ns, not significant. **, $P < 0.01$. Box plots are shown with minimum to maximum values, inner quartile ranges (box), median (line in box), mean (plus sign in box), and outer quartile ranges (whiskers).

on *B. cinerea* metabolism. We examined sugar uptake, glycolysis, and cellular redox status of *B. cinerea* in the presence of CK. Our results demonstrate that the inhibitory activity of CK against *B. cinerea* is largely dependent on the status of energy availability.

**Inhibition by CK is dependent on sugar availability to the fungus.** We found that the inhibitory effect of CK on *B. cinerea* was correlated with sugar availability. *B. cinerea* grown in defined or PDA medium with adequate sugar levels were inhibited by CK, while the growth of *B. cinerea* in sugar-depleted defined medium was promoted by CK (Fig. 1). In diluted rich medium, the inhibitory effect of CK diminished with medium dilution (Fig. S1 and S2). Intact F-actin was found to be required for hyphal growth, morphogenesis, and virulence, which were all impaired in F-actin capping protein deletion mutants (54). Since we had previously observed that CK caused mislocalization of actin at the growing tip of *B. cinerea* hyphae (38), we examined whether this phenomenon was also correlated with the nutritional status of the environment. We found that lowering nutrient availability results in deregulation of actin polarity that is required for growth, similarly to the effect of CK on actin distribution (Fig. 2),

suggesting that CK-dependent growth inhibition and growth inhibition as a result of nutrient-restriction may rely in part on similar mechanisms.

**CK promotes fungal metabolism.** Our previously published transcriptome profile revealed that important developmental pathways such as cell division, cellular trafficking, and the cytoskeleton, were inhibited upon CK treatment. Our results demonstrated that the effect of CK on *B. cinerea* is dependent on sugar availability. Reexamining our transcriptomic data in light of this, we found that glycolysis, sucrose metabolism, and oxidative phosphorylation pathways were significantly enriched upon CK treatment (Fig. 3). The expression of sugar transporters was also significantly upregulated.

This transcriptomic data were generated under one set of nutrient conditions. To further examine possible effects of CK on fungal metabolism, we investigated the effect of CK on *B. cinerea* glucose utilization and ATP synthesis, under different nutrient and energy availability conditions. We found that CK promoted an increase in sugar utilization by the fungus, in a sugar-availability-dependent manner, with the strongest promotion observed under sugar restriction (Fig. 5; Fig. S5). Interestingly, mineral element restriction, abolished this CK-dependent sugar utilization increase, suggesting that these elements are required for CK-mediated promotion of sugar utilization. Further work is needed to investigate the role of different mineral elements in CK activity in fungi. Growth attenuation by glycolysis inhibition using 2-DG occurred under glucose restriction, but not in full medium or in mineral element restriction (Fig. 4). CK was able to rescue inhibition of glycolysis and ATP synthesis under glucose restriction in defined media (Fig. 4) or upon PDA medium dilution (Fig. S6). This suggests that the results observed in PDA medium dilution are primarily due to the dilution of the sugar source (dextrose). Upregulation of glycolysis and oxidative phosphorylation key genes, together with the increased utilization of sugar, could explain the rescue of metabolic inhibition by CK. Taken together, these results confirm that the effect of CK on *B. cinerea* is dependent on sugar availability. Further work is needed to examine whether the increases in sugar utilization upon CK treatment reflect a true increase in glycolytic rates, but the fact that this increase occurs in both rich and sugar-restricted defined media (although at different levels) does seem to suggest that this might be the case.

**CK alters fungal redox.** Changes in metabolic pathways are often reflected in the redox status (50). Hence, we examined the effect of CK on *B. cinerea* redox status both *in vitro* and *in planta*, using tomato genotypes with various CK content or sensitivity. *B. cinerea* cells grown with CK in rich medium had a significantly reduced cytosol and a significantly oxidized mitochondria (Fig. 6). A reduced cytosol and oxidized mitochondria is indicative of increased glycolysis and oxidative phosphorylation (55, 56). Thus, this result correlated with our transcriptomic data, in which glycolysis and oxidative phosphorylation are upregulated in the presence of CK (Fig. 3), and also with our results demonstrating that CK promotes sugar utilization in *B. cinerea* (Fig. 5). *In planta*, after 48 h of inoculation, *B. cinerea* had a reduced cytosol and oxidized mitochondria when infecting the CK-rich IPT and an oxidized cytosol and reduced mitochondria when infecting the CK-deficient CKX, confirming that CK can affect the redox state of *B. cinerea* during infection *in planta* (Fig. 7; Fig. S6 and S7). This significant change in redox state was also coupled with the lower virulence on IPT leaves and higher virulence on CKX leaves (Fig. S10) (4). It was previously reported that the cytosol of infecting *B. cinerea* hyphae on dead onion peel is reduced (52). Our results also show that the *B. cinerea* cytosol is more reduced when infecting IPT overexpressing leaves, but the resultant infection was lower compared to that on CKX. In addition to the different hosts systems and different infection time frames, a possible explanation for this could be the CK-mediated immunity induced in the host (4, 12). The results concerning the redox state of *B. cinerea* hyphae *in vitro* and *in planta*, together with transcriptome data and sugar utilization results, might hint at the role of CK in nutrient allocation in fungi during infection.

**CK defines and supports tissues for fungal use.** CKs have previously described roles in plant-pathogen interactions (3, 4). The interaction of some biotrophic pathogens with their hosts leads to the formation of green bionissia (formerly known as

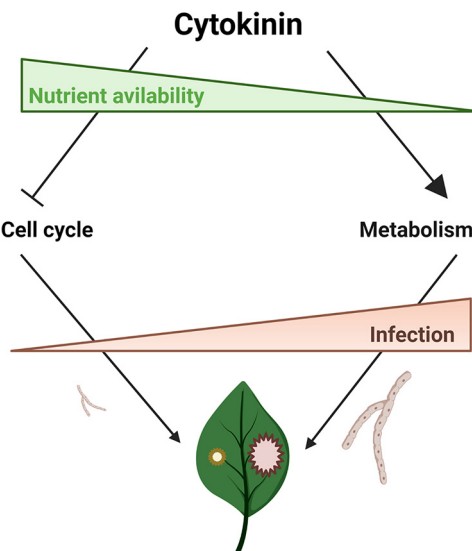

**FIG 9** Graphic model summarizing the posited dual role of CK in plant-fungi interactions. CK can signal for both growth inhibition or metabolic increases in the fungus, dependent on sugar availability. In high nutritional availability, the growth inhibitory effects are more dominant, with CK causing cell cycle arrest in the fungus, resulting in less fungal growth and therefore weaker plant infection. Under energy-restrictive conditions, the positive metabolic effects are more dominant, leading ultimately to stronger plant infection. The model was created with BioRender.com.

green islands) (57), which are sites of green living tissue surrounding the sites of active pathogen growth (9). The formation of these green bionissia is correlated with elevated levels of cytokinins in these tissues. It is believed that cytokinins likely delay the onset of senescence in green bionissia, allowing pathogens access to more nutrients from the plant (9). Necrotrophic fungal pathogens, which obtain their nutrients from dead plant cells, have also been reported to cause the formation of green tissue around the sites of infection (green necronissia) in certain cases (9). Application of exogenous CK is known to induce the formation green necronissia (9, 15, 58). We found that the effect of CK on *B. cinerea* is dependent on nutrient availability and can induce glycolysis, oxidative phosphorylation, and sugar utilization. We also observed CK-mediated redox state shifts in *B. cinerea* that are likely due to these increases in metabolism.

We previously observed that CK inhibits *B. cinerea* growth and development *in vitro*, in plant-physiological concentrations. The metabolic effects of CK on the fungus likely reflect the role of plant CK during early infection by necrotrophic pathogens, which have been demonstrated to have a short biotrophic phase (23, 59). Necrotrophic pathogens secrete toxins and enzymes to cause cellular damage. Thus, in addition to the plant producing CK during its interaction with the pathogen for the purpose of priming its defenses and inhibiting pathogen growth, the pathogen may take advantage of this increase in CK to exploit the formation of green necronissia, by increasing its metabolism and energy production to prepare for the necrotrophic phase of the infection. Thus, the role of CK in controlling senescence, which is used by biotrophic pathogens to their advantage, likely also benefits necrotrophs during their biotrophic phase. However, when CK content is high, as in IPT, the CK is also directly inhibitory against the pathogen, causing an attenuation of virulence, rendering the advantage of the CK to the pathogen in this initial phase of the infection irrelevant.

**CK has a dual role in plant-fungal interactions.** CK essentially provides two different cues for the fungus and is able to signal for both growth inhibition, to the benefit of the plant, and increased energy utilization, to the benefit of the fungus. Fig. 9 depicts a model graphically summarizing the proposed dual role of CK in this interaction. Our work demonstrates that under high nutritional status, the growth inhibitory effects are more dominant, while under low nutrient status, when the fungus is growing more slowly, the positive metabolic effects are more dominant (Fig. 9). These observations could explain, in

part, why *B. cinerea* is better able infect older plants (60–64), and also why it is sometimes customary to grow fungi in minimal medium prior to plant infection. This leads to the likely conclusion that one of the reasons why a starving fungus is more virulent is that it is less sensitive to inhibitory signals from the plant.

Plant-pathogen interactions involve signals from one partner being interpreted and responded to by the other partner, with each side tailoring its output to inputs received from the other side. Under low nutritional status, such as a plant leaf surface or intercellular spaces, a fungal spore that lands and is "searching" for a food source is growing very slowly and therefore is not inhibited, or is perhaps even promoted, by CK (Figs. 1 and 9; Fig. S1). Tomato leaves contain 100 to 450 nM *trans*-zeatin and additional amounts of other CKs, several of which have been found to affect *B. cinerea* (38). CK present on the leaf surface or intercellular spaces at low levels could serve as a cue for the fungus to increase its growth and metabolism in order to improve its ability to penetrate and grow massively within the plant to achieve successful infection and lyse the cells to utilize the available nutrients. Under low nutrition, CK is therefore a cue to the fungus: "Get ready. There will soon be high nutrients and a limited time to exploit them." Under higher fungal loads, when cells have been lysed and cellular content is leaked, the fungus then encounters 100 to 500 nM CK (38). This is still not a highly inhibitory level (38), and furthermore, the amount of CK decreases as infection starts (38). However, 48 h after fungal exposure, the plant starts making higher levels of CK (38, 65). These higher levels can now be inhibitory to the fungus, particularly as the environment within the plant is nutrient rich, and therefore, CK becomes more inhibitory (Figs. 1 and 9; Fig. S1). The polarity of F-actin is likely indicative of growth, as in minimal medium; even without CK, this polarity is reduced (Fig. 2). CK reversibly inhibits the cell cycle of *B. cinerea* (38). Here, we decipher another piece of the puzzle, showing that CK can also have positive effects as a signal to the fungus to increase energy metabolism under low nutrient status, cementing the role of CK as a bidirectional signaling molecule in plant-fungal interactions.

Our work suggests that CK may serve as central player in the hormonal cross-talk between plant host and phytopathogen, and that the role of CK in controlling senescence can be exploited by diverse fungal phytopathogens to their advantage. Future research will focus on the role of CK in nutrient allocation in fungal phytopathogens during infection, affording insights into fungal infection phases in the context of host-phytopathogen interactions.

## MATERIALS AND METHODS

**Fungal growth conditions.** *B. cinerea* was grown in different types of defined liquid media or different strengths of PDA or PDB, as detailed in Table 1, at $22 \pm 2°C$, in the dark, with 150 rpm shaking for liquid media. *B. cinerea* was grown for 6 to 72 h in liquid PDB or 5 days on solid PDA.

*B. cinerea* strains were obtained from Yigal Elad from the following hosts: cucumber (Bcl-16, Bcl-C1, Bcl-C3), eggplant (Bcl-E1, Bcl-E2), grapevine (Bcl-V1, Bcl-V2, Bcl-V3, Bcl-V4, Bcl-V5, Bcl-V6), pepper (Bcl-P1, Bcl-P2), strawberry (Bcl-S1, Bcl-S2), and tomato (Bcl-T1). The lab strain B05.10, originally from grapevine, was also used. Most of the work was done with Bcl-16, which sporulates well on different types of media, including PDA (38).

**Mycelial growth assay.** To study how nutrient availability affects growth of *B. cinerea* in the presence of CK, 6-benzylaminopurine (6-BAP; 100 $\mu$M, Sigma-Aldrich B3408) was dissolved in 10 mM NaOH and added to the different media detailed in Table 1. To study how *B. cinerea* responds to metabolic inhibitors in the presence of CK, 2-deoxy-D-glucose (2-DG; Sigma-Aldrich) and oligomycin (OM; Sigma-Aldrich) were used. 2-DG was added to the different media detailed in Table 1 above at a final concentration of 2.5 mM. Oligomycin (OM), which proved too inhibitory in defined media, was assayed in solid PDA alone at 0.1 $\mu$g/mL. For liquid medium, *B. cinerea* spores ($10^6$/mL) were grown in defined medium or PDB for 72 h, after which the fungal mass was dried, and the dry weight was measured. For PDA plates, mycelial plugs (5 mm) taken at ~1 cm from the edge of a fresh plate were placed at the center of PDA plates and incubated at $22 \pm 2°C$ in the dark for 5 days.

**Glucose utilization.** To evaluate the effect of CK on glucose utilization, the decline in glucose levels in the medium following fungal growth was measured. *B. cinerea* conidia were harvested in 1 mg mL$^{-1}$ glucose and 1 mg mL$^{-1}$ K$_2$HPO$_4$ and filtered through a 40-$\mu$m pore cell strainer (Corning). Spore concentration was adjusted to $10^6$ spores mL$^{-1}$ using a Neubauer chamber. *B. cinerea* spores were grown in liquid media: PDB or defined media as detailed in Table 1 above. A total of 100 $\mu$M CK was added to both PDB and defined media cultures, which were then allowed to grow for 48 h. The amount of metabolized glucose was analyzed by measuring the amount of glucose present in the media by the 3, 5

**TABLE 1** Specifications of different media types used in this work

| Name | Sugar source | Nutrient source | Manufacturer |
|---|---|---|---|
| Defined medium 1 | 20 g/liter glucose | 4 g/liter each of $K_2HPO_4$, $KH_2PO_4$, and $(NH_4)_2SO_4$ | Sigma-Aldrich (components) |
| Defined medium 2 | 5 g/liter glucose | 4 g/liter each of $K_2HPO_4$, $KH_2PO_4$, and $(NH_4)_2SO_4$ | Sigma-Aldrich (components) |
| Defined medium 3 | 20 g/liter glucose | 1 g/liter each of $K_2HPO_4$, $KH_2PO_4$, and $(NH_4)_2SO_4$ | Sigma-Aldrich (components) |
| Defined medium 4 | 5 g/liter glucose | 1 g/liter each of $K_2HPO_4$, $KH_2PO_4$, and $(NH_4)_2SO_4$ | Sigma-Aldrich (components) |
| PDA/PDB | 20 g/liter dextrose | 4 g/liter potato starch | Difco (39 g/liter) |
| 1/2 PDA/PDB | 10 g/liter dextrose | 2 g/liter potato starch | Difco (19.5 g/liter) |
| 1/4 PDA/PDB | 5 g/liter dextrose | 1 g/liter potato starch | Difco (9.75 g/liter) |
| 1/8 PDA/PDB | 2.5 g/liter dextrose | 0.5 g/liter potato starch | Difco (4.875 g/liter) |

Dinitrosalicylic acid (DNSA) method (66), using dextrose as a standard. 3,5-Dinitrosalicylic acid was purchased from Sigma-Aldrich. The utilized glucose was assumed to be inversely proportional to the amount present in the medium. For control, glucose was measured in the different media, after they were subjected to the above-mentioned conditions, without *B. cinerea*.

**Generation of transgenic *B. cinerea* lines.** For generation of *B. cinerea* strains expressing Lifeact-GFP, we used a fusion construct to target replacement of the nitrate reductase (bcniaD) gene, as reported previously by our group and others (38, 39). The plasmid pNDH-OLGG, which was used as a template for the amplification of expression cassette, has flanking sequences of bcniaD, a resistance cassette for hygromycin, and the filamentous actin (F-actin) imaging probe "Lifeact" fused to GFP. Primers GA 34F/34R (Table S1) from our previous study (38) were used for the amplification of the expression cassette. Polyethylene glycol (PEG)-mediated transformation was used to transfer the PCR-amplified expression cassette to *B. cinerea* (38, 67). Fungal transformants were visualized under a confocal microscope and screened with primers GA 44F/44R and GA 31F/31R (Table S1). To examine the effect of CK on the cytoskeleton under different sugar and nutrient levels, spores of transformed *B. cinerea* were treated with mock or CK (100 $\mu$M) in full and 1/4 PDB and grown for 6 and 24 h, respectively, prior to confocal visualization. The Lifeact-GFP transformed fungus has abnormal growth, as previously reported (38, 39) and could not be grown in the defined media detailed in Table 1 above.

For the detection of redox state in cytosol and mitochondria, *B. cinerea* expressing GRX-roGFP and mito-roGFP were generated. For the expression of the redox sensors, a construct generated previously was used (39, 51, 52). For generation of constructs expressing GRX-roGFP at the bcniaA locus, we used the plasmid pNAH-GRX-roGFP$^{CYT}$ as the template. The vector contains 5'- and 3'-flanking sequences of bcniaA, a resistance cassette mediating resistance to hygromycin, and the sensor for the redox potential of the cellular glutathione pool glutaredoxin probe "GRX" fused to GFP. The cassette carrying the hygromycin resistance gene, the expression cassette for GRX-roGFP/mito-roGFP, and the bcniaA flanking sequence were amplified using primers GA 41F/41R (Table S1). For the amplification of mito-roGFP, the plasmid pNAH-roGFP$^{MITO}$ was used as a template. PEG-mediated transformation was used to transfer the PCR-amplified expression cassette to *B. cinerea* (67). Fungal transformants were visualized under a confocal microscope and screened with primers GA 42R/42R (Table S1).

**PEG-mediated *B. cinerea* transformation.** PCR-amplified expression cassettes as detailed above were used to transform *B. cinerea* using PEG-mediated transformation. 0.125% lysing enzyme from *Trichoderma harzianum* (Sigma-Aldrich, Germany) was used for protoplast generation. Following PEG-mediated transformation, protoplasts were plated on SH medium containing sucrose, Tris-Cl, $(NH_4)_2HPO_4$, and 35 $\mu$g/mL hygromycin B (Sigma-Aldrich, Germany). Colonies that grew after 2 days of incubation were transferred to PDA-hygromycin medium, and conidia were spread again on selection plates to obtain a monoconidial culture. Fungal transformants were confirmed by confocal microscopy and PCR screening, as detailed for each construct. Confirmed transformants were stored at −80°C and used for further experiments.

**Confocal microscopy.** We acquired confocal microscopy images using a Olympus IX 81 inverted laser scanning confocal microscope (Fluoview 500) equipped with an OBIS 488-nm laser and a 60 × 1.0 NA PlanApo water immersion objective. GFP images of 24 bits and 1,024 × 1,024 pixels were acquired using the excitation/emission filters: BP460-480GFP/BA495-540GFP. Image analysis was conducted with Fiji-ImageJ using the raw images, the 3D object counter tool, and the measurement analysis tool (68).

**Transcriptome analysis of metabolic pathway genes.** The transcriptomic analyses presented in this work were carried out on previous data, first published by Gupta et al. (38). Procedures for RNA preparation, quality control, sequencing, and transcriptome analysis are detailed in our previous work (38). RNA preparation included a DNase (Sigma-Aldrich) treatment step. Differential expression analysis was executed using the DESeq2 R package (69). Genes with an adjusted *P* value of no more than 0.05 were considered differentially expressed. PCA was calculated using the R function prcomp. The sequencing data generated in this project was previously published (38), and the raw data are available at NCBI under Bioproject accession number PRJNA718329.

The gene sequences were used as a query term for a search of the NCBI nonredundant (nr) protein database that was carried out with the DIAMOND program (70). The search results were imported into Blast2GO version 4.0 (71) for Gene Ontology (GO) assignments. Gene Ontology enrichment analysis was carried out using Blast2GO program based on Fisher's exact test with multiple testing correction of false discovery rate (FDR). The KEGG pathway database (https://www.genome.jp/kegg/) (72) was used to detect the statistical enrichment of differential expression genes in KEGG pathway.

**RNA preparation and qRT-PCR.** To examine the effect of cytokinin on fungal glycolysis and validate the RNA-seq results, we grew *B. cinerea* spores in PDB with 0 and 100 $\mu$M 6-BAP in a rotary shaker at 150 rpm and 22 $\pm$ 2°C for 24 h. Total RNA was isolated using Tri reagent (Sigma-Aldrich) according to the manufacturer's instructions. RNA preparation included a DNase (Sigma-Aldrich) treatment step. RNA (3 $\mu$g) was used to prepare cDNA using reverse transcriptase (Promega, USA) and oligo(dT)15. qRT-PCR was performed on a Step One Plus real-time PCR system (Thermo Fisher, Waltham, MA, USA) with Power SYBR green Master Mix protocol (Life Technologies, Thermo Fisher, USA). For glycolysis analysis, we selected the following genes: glyceraldehyde-3-phosphate dehydrogenase (XP_024553281.1), pyruvate dehydrogenase (XP_001558781.1), aldehyde dehydrogenase (XP_001554714.1), and alcohol dehydrogenase (XP_001554746.1). For purine permease (PUP) expression analysis, we selected purine nucleoside permease (*PUP1*, XP_024546421.1), purine-cytosine permease (*PUP2*, XP_001559974.1), and purine nucleoside permease (*PUP3*, XP_024548226.1). The primer sequences for each gene, as well as primer pair efficiencies, are detailed in Table S2. A geometric mean of the expression values of the three house-keeping genes: ubiquitin-conjugating enzyme E2 (ubce) (73), iron-transport multicopper oxidase, and adenosine deaminase (74) was used for normalization of gene expression levels. All primer efficiencies were in the range 0.97 to 1.03 (Table S2). Relative expression was calculated using the copy number method (75). At least five independent biological replicates were used for analysis.

**Measurement of *B. cinerea* redox status in liquid medium.** Redox state of roGFP transformed *B. cinerea* was measured in liquid culture using a fluorimeter. *B. cinerea* strains expressing GRX-roGFP and mito-roGFP were grown for 2 weeks on PDA medium at 18°C in the light to induce mass sporulation. A total of 10 mL of PDB medium containing 100 $\mu$M CK was inoculated with conidia and incubated for 24 h at 18°C, with 150 rpm shaking. Samples of 1 mL were taken and washed twice in double-distilled water, and 200 $\mu$L of the washed germlings were transferred to a 96-well plate (Microplate pureGrade 96-well PS, transparent bottom) for fluorescence measurements using a fluorimeter (Promega GloMax explorer multimode microplate reader, GM3500, USA). Fluorescence was measured at the bottom with 3 $\times$ 3 reads/well and an excitation wavelength of 405 $\pm$ 5 nm for the oxidized state and 488 $\pm$ 5 nm for the reduced state of roGFP2. The emission was detected at 510 $\pm$ 5 nm (52). The gain was set to 100. Relative fluorescence units (RFU) were recorded to calculate the $Em_{405}/Em_{488}$ ratio.

**Imaging of *B. cinerea* redox state during on-plant pathogenesis.** To examine the effect of endogenous CK content on the redox state of *B. cinerea* during pathogenesis, *B. cinerea* GRX-roGFP conidia from freshly sporulated PDA plates were used to infect detached leaves from the *S. lycopersicum* M82 background line, as well as M82 overexpressing the *Arabidopsis* isopentenyl transferase (IPT) gene *AtIPT7* under the FIL promoter: *pFIL $\gg$ IPT7* (IPT), and M82 overexpressing the *Arabidopsis* cytokinin oxidase (CKX) gene *AtIPT3* under the BLS promoter: *pBLS $\gg$ CKX3* (CKX) (4). For inoculation, *B. cinerea* was grown on PDA in the dark at 22 $\pm$ 2°C. After 6 h of daylight, 10-day-old plates were returned to the dark for sporulation. The spores were harvested in 1 mg mL$^{-1}$ glucose and 1 mg mL$^{-1}$ K$_2$HPO$_4$ and filtered through a 40-$\mu$m pore cell strainer (Corning). Spore concentration was adjusted to $10^6$ spores mL$^{-1}$ after quantification under a light microscope using a Neubauer chamber. Leaflets from the third leaf of 15-day-old tomato plants were excised and immediately placed in humid chambers. Leaflets were inoculated with one droplet of 5-$\mu$L spore suspension. Twenty-four hours after inoculation, germinated conidia were imaged using a fluorescent Olympus IX 81 inverted laser scanning confocal microscope (Fluoview 500), as detailed above. Images were collected with a 60 $\times$ 1.0 NA PlanApo water-immersion lens in multitrack line mode. roGFP was excited at 405 nm in the first track and at 488 nm in the second track. For both excitation wavelengths, roGFP fluorescence was collected with a bandpass filter of 505 to 530 nm and averaged from four readings for noise reduction (52). The ratiometric analyses of fluorescence images were calculated using Fiji-ImageJ.

**CK internalization.** *B. cinerea* conidia were harvested in 1 mg mL$^{-1}$ glucose and 1 mg mL$^{-1}$ K$_2$HPO$_4$ and filtered through a 40-$\mu$m pore cell strainer (Corning). Spore concentration was adjusted to $10^6$ spores mL$^{-1}$ using a Neubauer chamber. *B. cinerea* spores were grown in liquid PDB, at full strength for 6 h, or 1/4 strength for 24 h. 10 $\mu$M iP-NBD was then added to the germlings, which were grown with the fluorescent CK for an additional 2 h. The fungal matter was then washed in water, resuspended in 1 mg mL$^{-1}$ glucose and 1 mg mL$^{-1}$ K$_2$HPO$_4$, and mounted for confocal microscopy as detailed above, using the GFP track. Internalized CK was quantified from 17 images per treatment, using Fiji-ImageJ. Mean fluorescence was calculated using the measure tool, and the corrected total cell fluorescence was calculated as the Integrated Density – (Area of selected cell X Mean fluorescence of background readings).

**Data analysis.** The data are presented as minimum to maximum values in box plots or as averages $\pm$ SEM in bar graphs. For Gaussian distributed samples, we analyzed the statistical significance of differences between two groups using a two-tailed *t* test, with additional *post hoc* correction where appropriate, such as Welch's correction for *t* tests between samples with unequal variances. We analyzed the statistical significance of differences among three or more groups using analysis of variance (ANOVA). Regular ANOVA was used for groups with equal variances, and Welch's ANOVA for groups with unequal variances. Significance in differences between the means of different samples in a group of three or more samples was assessed using a *post hoc* test. Tukey's *post hoc* test was used for samples with equal variances, when the mean of each sample was compared to the mean of every other sample. Bonferroni's *post hoc* test was used for samples with equal variances, when the mean of each sample was compared to the mean of a control sample. Dunnett's *post hoc* test was used for samples with unequal variances. For samples with non-Gaussian distribution, we analyzed the statistical significance of differences between two groups using a Mann-Whitney U test, and the statistical significance of differences among three or more groups using Kruskal-Wallis ANOVA, with Dunn's multiple comparison *post hoc* test as indicated. Gaussian distribution or the lack thereof was determined using the Shapiro-Wilk test for normality. Statistical analyses were conducted using Prism8.

**Data availability.** All of the data supporting the findings of this study are available within the article and its supplementary information files. The raw data are available from the corresponding author upon reasonable request. The raw data generated in the previously published transcriptomic analyses have been deposited in NCBI under Bioproject accession number PRJNA718329.

## SUPPLEMENTAL MATERIAL

Supplemental material is available online only.
**SUPPLEMENTAL FILE 1**, PDF file, 0.9 MB.
**SUPPLEMENTAL FILE 2**, XLSX file, 0.05 MB.

## ACKNOWLEDGMENTS

This work was supported by the Israel Science Foundation grant 1759/20 (to M.B.). The funder had no role in study design, data collection and analysis, decision to publish, or preparation of the manuscript. G.A. is partially supported by the Indo-China ARO Postdoctoral Fellowship Program.

The Lifeact and roGFP constructs were kindly provided by Julia Schumacher. We thank Ondrej Plihal and colleagues for providing iP-NBD. We thank members of the Bar group for continuous discussion and support.

Conceptualization: G.A., M.B. Design: G.A., R.G., M.B. Methodology & experimentation: G.A., R.G. Analysis: G.A., R.G., M.B. Manuscript: G.A., M.B.

We declare no conflict of interest.

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
