## [Reviewer comments · Microbiology Spectrum]

Microbiology Spectrum

Cytokinin regulates energy utilization in *Botrytis cinerea*

Gautam Anand, Rupali Gupta, and Maya Bar

Corresponding Author(s): Maya Bar, Agricultural Research Organization

Review Timeline:

Submission Date:	January 23, 2022
Editorial Decision:	March 5, 2022
Revision Received:	April 29, 2022
Editorial Decision:	May 23, 2022
Revision Received:	July 9, 2022
Accepted:	July 11, 2022

Editor: Giuseppe Ianiri

Reviewer(s): The reviewers have opted to remain anonymous.

Transaction Report:

DOI: <https://doi.org/10.1128/spectrum.00280-22>

Dr. Maya Bar
Agricultural Research Organization
Plant Pathology and Weed Research
Bet Dagan
Israel

Re: Spectrum00280-22 (Cytokinin regulates energy utilization in *Botrytis cinerea*)

Dear Dr. Maya Bar:

I have received the reviews of your manuscript entitled "Cytokinin regulates energy utilization in *Botrytis cinerea*", and I regret to inform you that we will not be able to publish it in Spectrum. Your submission was read by reviewers with expertise in the area addressed in your study and it was the consensus view of these reviewers that your paper did not meet the standards necessary for publication. In particular, all reviewers criticized the experimental set up that was based on the use of a undefined media (PDA), which does not allow the claimed correlation between nutrients and CK activity. Copies of the reviewers' comments are attached for your consideration.

I am sorry to convey a negative decision on this occasion, but I hope that the enclosed reviews are useful. Please note, rejections from Microbiology Spectrum are final and your manuscript will not be considered by other ASM journals. We wish you well in publishing this report in another journal and hope that you will consider Spectrum in the future.

Sincerely,

Giuseppe Ianiri
Editor, Microbiology Spectrum

Reviewer comments:

Reviewer #1 (Comments for the Author):

In this manuscript, a wealth of experiments have been carried out on the possible dual effects of cytokinin on *Botrytis cinerea*, and the research angle is very novel, which is promising to find the key signaling substances when Bc infects plants. However, the writing of this manuscript has obvious defects:

1, Cytokinins are plant-specific chemical messengers (hormones) that play a central role in the regulation of the plant cell cycle and numerous developmental processes (T. Schmülling, in Encyclopedia of Biological Chemistry (Second Edition), 2013). This article does not specify anywhere which cytokinin is, and does not provide a molecular formula or commodity number of the cytokinin.

2, Regarding the experimental design based on PDA, I personally have some doubts. First, the author did not specify the preparation method or manufacturer of PDA; second, the composition of PDA is complex, and precipitation may occur after sterilization, and there are interference factors when explaining the effect of CK; third, even in the context of PDA, when the nutrient composition is reduced to 1/4 and 1/8, the bacterial growth will be severely inhibited, the growth rate and metabolic level will be significantly reduced without CK treatment, and the individual believes that there is a minimum threshold for the reduction of its physiological level, that is to say, the weakening of the inhibitory effect of CK treatment may be caused by the impending touch of the minimum threshold, and the intensity of CK active regulation is difficult to directly prove in this study.

3, There are many problems in the description of methods in this manuscript, such as unknown source of strains, unknown source of reagents and instruments, and lack of corresponding references, and the specific problems have been labeled in the manuscript.

4, The discussion section did not discuss enough relevant research within the field. Because of completely new research findings?

5, There are still many missing information in this manuscript, which have been labeled in the manuscript .

Reviewer #2 (Comments for the Author):

Dear Authors,

First let me start with, I find the concept and the possibility that CK's influence metabolic functions in Bc depending on the nutrient (glucose) status of the environment intriguing.

Unfortunately, I have major concerns about the methods used and thus about the resulting data. If the authors can show that the observed effects are repeatable in a defined minimal medium with constant salt concentrations but variable glucose concentrations, I would be happy to reconsider my opinion.

Here are the main reasons, why I can't give a positive review:

In my opinion the usage of an undefined medium such as PDB or PDA and reducing its strength by dilution leads to changes of too many variables (glucose, nitrogen, phosphate, and many more) at the same time, which makes it impossible to directly link the observed phenotypes to glucose (which is the main statement of the manuscript). The one time a defined minimal medium was used it again was diluted, thus diluting glucose amounts, but also reducing the amounts of important minerals/salts such as potassium, phosphate, sulfur and nitrogen. Thus, the data interpretation is very difficult. I'm sure lower concentrations of nitrogen and especially phosphate will also influence glycolysis or mitochondrial functions related to energy production for example.

Please repeat everything with a defined minimal medium where only the glucose concentration is changed and the salt concentrations are kept constant. Or rewrite the manuscript with a reduced focus on glucose (which would devalue the information of the manuscript).

Other points: why is sometimes boxplots with whiskers used other times only boxplots without whiskers. Are the data shown as median (not a scientific common way to show data) as state most of the times or as averages? Why are standard deviations missing for example Fig 7 B and C? Why are sometimes the individual data points included other times not?

Fig 1: please redo with minimal medium where only the glucose concentration is changed.

Fig 2: is 6h or 24h shown? Or is one timepoint missing from the graph?

Fig 3: I find this correlation of environmental Bc strains with somewhat different growth rates and the influence of CK not very strong. There could be a multitude of reasons for those strains to behave different, which are not directly linked to CK. The authors should have used mutants in B05.10 or T4 for glucose related functions, uptake, glycolysis, TCA, ETC.. to show the impact of CK on glucose related functions (Best scenario). If this is not possible other strains with reduced growth rate, but with defined known mutations could be used as well. In Figure B the R2 is missing, which would show the fitting of the curve with the data. If the one high value would be missing, the line would be pretty flat and the resulting correlation would not be significant.

Fig 4: (B) I'm confused about the p value for either the PCR or RNAseq part of the graph. Is it for the whole group of genes as an average or? Please mark the significance for each individual gene. Further for the RNAseq results what is the unit of the y-axes? Is it log or relative regular fold change? If it is relative, why are the changes not even close to be similar to the RT PCR results, where are the standard deviations? I agree the pattern is similar between the two analyses, but the fold changes are not similar. Further was the RNA DNase treated?

Fig 5: Is it again mean or average, no whiskers,...? The recorded data and the resulting shown pattern of 1/4 media with DG is similar to the other pattern with higher media strength, but the spread of the data especially of DG and DG+CK make any interpretation even with statistics very difficult. In my opinion not a strong indicator that DG has an influence on CK function

Fig 6: For this kind of analysis and the conclusion that CK leads to differences in glucose uptake only glucose uptake studies with labelled glucose should be used. There could be many reasons why glucose is less abundant in the CK treated condition, higher binding capacity of glucose in the fungal cell wall, by extracellular proteins or formation of a glucose containing product, which can't be measured with the reducing sugar assay,... Best to use intact protoplasts for the uptake assay with labelled glucose.

Fig 7: Again median? Why no standard deviations in B and C? What is the number in C at 4h. Further, especially the differences in C at 24h are so small, that they are barely visible, which questions the biological relevance.

Fig 8: median? Whiskers?

Reviewer #3 (Comments for the Author):

The manuscript is a follow up on an earlier work in which the authors demonstrated that external application of the cytokinin 6-Benzylaminopurine (CK) can inhibit fungal growth. In the current study they used *Botrytis cinerea* to investigate the possible effect of CK on fungal metabolism and nutrition in vitro and on plants. They report that along with growth inhibition, CK promoted fungal metabolism and that these effects were inversely correlated with nutrient and sugar availability. Based on these findings, the authors propose that CK acts as "a signal to the fungus that plant tissue is present, causing it to activate energy metabolism pathways to take advantage of the available food source, while at the same time, CK is employed by the plant to inhibit the attacking pathogen". This is an intriguing concept with potential to highlight a largely ignored direction in plant microbe interactions.

While the phenomenon is interesting, there are many open questions that need to be answered. For example, how does the fungus sense CK while on the leaf? If only after plant penetration, then CK might not affect the early stages that include attachment, germination and penetration. What are the levels of CK in different types of tissues, in particular fruits and flowers, which are the most sensitive organs? Is CK evenly distributed throughout the tissues, how does (if it does?) the fungus sense it and how does it enter the fungal cell? These and similar questions warrant further research to validate the new concept. Additionally, there are some technical aspects that should be examined and it reflects on the interpretation of some results as specified below.

Overall, this is an intriguing paper, but more work is needed to validate the new concept and part of the methods and results need careful evaluation.

Main comments

Lines 28-31: "indicating that nutrient availability is indeed the switch determining the role of CK. Transcriptomic data further support our findings, demonstrating significant upregulation to glycolysis, oxidative phosphorylation, and sucrose metabolism, upon CK treatment. Thus, the effect of CK in fungal biology depends on energy status". This statement concludes that nutrient availability and specifically energy level determine CK effects. At present the data are insufficient to draw this conclusion since they are all circumstantial, and in some cases might not support this statement. This and similar statements and conclusions throughout the manuscript should be avoided, or at least drastically toned down.

Use of PDA: the central point of this work is possible connection between energy (carbon) levels and CK activity. PDA is an undefined medium (an extract of potatoes). It contains a wide range of metabolites other than C and N, for example amino acids, vitamins, elements etc. Each of these molecules can potentially affect CK activity in a range of ways. Therefore, the choice of PDA as the main nutrient medium is not optimal. It will be necessary to verify the phenomenon on defined media, such as CD, GH, etc., with different types of sugars.

Line 141 and similar/ Figure 1: "CK mediated growth inhibition is affected by nutrient availability". I am not convinced that this statement is correct. The data in Figure 1 show that the main effect on growth is medium concentration. The way the data are presented overlooks this fact. It is hard to calculate percentage based on the graphs, but by a rough estimate it seems that when expressed as % of control, CK has similar effects (around 50% inhibition) up to 1/4 strength. At 1/8 strength the growth is already minimal and the CK effect is therefore marginal. Additionally, as already suggested, these experiments should also be conducted on defined media.

Additional comments Figure 1: 1) Figure title is not necessarily supported by the data, 2) Under normal conditions *B. cinerea* grows very uniformly with minimal variance; What is the reason for the very high variance of growth measurements within treatments? 3). What you really measure is radial growth. Better present radius values, the area only inflates the numbers and might be misleading. 4) Would be good to test additional CK concentrations.

Figure 2: 1) The growth period on full and 1/4 strength media differs considerably; this factor can have immense impact on results, possibly far greater than the effect of CK. 2) At 1/4 strength the wild type also shows a lot of un-oriented actin namely there is a strong effect of the medium. Additionally, the way actin is counted is not necessarily reflecting changes due to CK effect. 3) What about results in 1/2 PDB? 4) What was CK concentration?

Figure 3: 1) Correlation is positive, but weak and probably doesn't mean much.

Minor comments

Line 22: When referring to the scientific name of an organism it is customary to write the first letter of the genus and the full name of the species. Suggest replacing all *Bc* with *B. cinerea*

Lines 68-69: Delete: "CK was also found to enhance disease resistance to additional, non obligatory plant pathogens". It is repeated in the following sentence.

Lines 70-71: What do you mean by "endogenous application"?

Line 130-131: This is probably one of the strongest results, but it does not necessarily relate to the strength of the medium.

Lines 189-190: This might not be the correct interpretation: the fact that there was no further inhibition doesn't necessarily mean the fungus was insensitive to CK.

Line 417: What is the source of this medium? What is the N source?

Line 446: If you did not carry additional RNAseq experiment, please explicitly mention that analyses were performed on old data.

Figure 5: 1) Reasons for high variance are unclear. 2) The strong effect of OM might mask the CK effect. Would be good to test at lower OM concentrations.

Figure 6: 1) Which medium was used and what was the main carbon source? 2) Presentation of % (top) might be misleading since the actual values are very low.

**Cytokinin regulates energy utilization in *Botrytis cinerea***

Gautam Anand, Rupali Gupta, and Maya Bar*

Department of Plant Pathology and Weed Research, ARO, Volcani Institute, Rishon LeZion
7505101, Israel.

*Corresponding author: Maya Bar

mayabar@volcani.agri.gov.il

ORCID: 0000-0002-7823-9121

Short title: Cytokinin and *Botrytis* energy status

Keywords: *Botrytis cinerea*, Cytokinin, Pathogenesis, Nutrition, Glycolysis, Redox

**Abstract**

The plant hormone cytokinin (CK) is an important developmental regulator. Previous work
has demonstrated that CKs mediate plant immunity and disease resistance. Some
phytopathogens have been reported to secrete CKs, and may manipulate CK signaling to
improve their pathogenic abilities. In recent work, we demonstrated that CK directly inhibits
the development and virulence of fungal phytopathogens, by down regulating the cell cycle
and reducing cytoskeleton organization in the fungus. Here, focusing on *Botrytis cinerea*
(*Bc*), we report that CK possesses a dual role in fungal biology, with role prioritization being
based on nutrient availability. In a nutrient rich environment, CK strongly inhibited *Bc*
growth and de-regulated cytoskeleton organization. This effect diminished as nutrient
availability decreased. In its second role, we show using biochemical assays and transgenic
redox sensitive fungal lines, that CK can promote glycolysis and energy consumption in *Bc*,
both *in vitro* and *in planta*. Glycolysis and increased oxidation mediated by CK were stronger
with waning nutrient availability, indicating that nutrient availability is indeed the switch
determining the role of CK. Transcriptomic data further support our findings, demonstrating
significant upregulation to glycolysis, oxidative phosphorylation, and sucrose metabolism,
upon CK treatment. Thus, the effect of CK in fungal biology depends on energy status. In
addition to the plant producing CK during its interaction with the pathogen for defense
priming and pathogen inhibition, the pathogen may take advantage of this increased CK to
boost its metabolism and energy production, in preparation for the necrotrophic phase of
the infection.

**Importance**

The hormone cytokinin (CK) is a plant developmental regulator. Several works have
highlighted the involvement of CK in plant defense. Here, we report that CK has a dual role
in plant-fungus interactions, inhibiting fungal growth under high nutrient conditions while
positively regulating *B. cinerea* energy utilization, causing an increase in glycolytic rates and
energy consumption. The effect of CK on *B. cinerea* was dependent on nutrient availability,
with CK primarily causing increases in glycolysis when nutrient availability was low, and
growth inhibition in a high nutrient environment. We found this effect to be preserved
across 17 different *Bc* isolates from 6 different hosts. We propose that CK acts as a signal to
the fungus that plant tissue is present, causing it to activate energy metabolism pathways to
take advantage of the available food source, while at the same time, CK is employed by the
plant to inhibit the attacking pathogen.

**Introduction**

Plant cytokinins (CKs) are known to be important in many aspects of plant life,
including development of vasculature, differentiation of embryonic cells, seed development,
maintenance of meristematic cells, growth and branching of root, shoot formation,
chloroplast biogenesis, and leaf senescence (1, 2). CKs also play an important role in
nutrient balance and stress responses in the plant (3, 4). They are known to influence
macronutrient balance by regulating the expression of nitrate, phosphate and sulphate
transporters (5–8). In addition, roles for CKs in fungal pathogenesis have also been
suggested, either in the context of the pathogens producing CKs, or in the context of the
pathogen activating the CK pathway in the host plant (9–12). Jameson, (13) suggested that
to achieve pathogenesis in the host, CK-secreting fungal biotrophs or hemibiotrophs alter CK
signalling to regulate the host cell cycle and nutrient allocation. For instance, germinating
uredospores of *Puccinia spp.* have been shown to accumulate CK, modifying CK signalling to
maintain plant cell cycle (14, 15). Plant CK levels can be modulated by the application of
exogenous CKs, and several studies have found a positive effect of CK treatment in
reduction of diseases caused by smut fungi, powdery mildew, and viruses (10, 16–18). More
recently, CK was also found to enhance disease resistance to additional, non obligatory
plant pathogens. Elevated levels of CKs were shown to increase host resistance to various
pathogens in a wide range of plants (4, 12, 19–21). We recently reported that endogenous
and exogenous applications of CKs induces systemic immunity in tomato, enhancing
resistance against fungal pathogen *Botrytis cinerea* (*Bc*) by salicylic acid and ethylene
signalling (4).

*Bc*, the causative agent of grey mould disease, is a cosmopolitan pathogen that can
infect more than 1400 host plants, and causes massive losses worldwide annually (22). *Bc* is

a mostly (23, 24) necrotrophic pathogen which has been widely used as a model pathogen
to study various mechanism underlying plant-pathogen interactions. Due to its economic
importance, *Bc* is listed in top 10 important plant fungal pathogens (25). During
pathogenesis, *Bc* induces necrosis in the host by producing various toxins such as botrydial,
botcinic acid, and its derivatives (26, 27) and production of reactive oxygen species (ROS),
and also manipulates the host plant into generating oxidative bursts that facilitate
colonization (28, 29) and promote extension of macerated lesions by the induction of
apoptotic cell death. Various enzymes including lytic enzymes, which are sequentially
secreted by the fungus, facilitate penetration, colonization, and produce an important
source of nutrients for the fungus (30).

*Bc* spores are primary sources of infection to plants in nature. After contacting the
plant surface, the spores germinate to form short germ tubes that directly penetrate plant
tissues (31). It is known since long that germination of spores and infection through plant
surfaces relies on their ability to access the nutrient supply offered by living plants (32, 33).
Solomon and co-workers (34) have suggested a model that describes nutrient availability to
the fungi during different phases of fungal infection to plant host. The first phase, which
involves spore germination and host penetration, is based on lipolysis. The second phase,
which requires invasion of plant tissues, uses glycolysis. Studies on *Tapesia yellundae*,
*Colletotrichum lagenarium* and *Cladosporium fulvum* suggest that lipids are the main
sources of energy during germination and penetration and after penetration the available
plant sugars become the main source of energy (35–37). These different stages of fungal
infection and metabolism depend on nutrient availability and allocation. Role of CK in
nutrient balance in plant is well studied but its role in nutrient balance and metabolism in
necrotrophic fungus like *Bc* needs to be investigated.

Recently, we have found that CK inhibits the growth of a variety of plant pathogenic
fungi. In depth characterization of the phenomenon in botrytis revealed that CK in plant
physiological concentrations can inhibit sporulation, spore germination, and virulence (38)
of *Bc*. We also found CK to affect both budding and fission yeast. Transcriptome profiling of
*Bc* grown with CK revealed that DNA replication and the cell cycle, cytoskeleton integrity,
and endocytosis, are all repressed by CK (38).

Given that CK had such fundamental, conserved, and ubiquitous effects on fungal
development, the question of a possible role for CK in affecting fungal metabolism and
nutrition, in particular during host-pathogen interactions, arises. In this study, we
investigated the effect of CK on fungal metabolism. Using *Bc*, we examined how CK affects
fungal metabolism and nutrition, both in the context of fungal growth and during infection
in the tomato host. We found that CK promoted fungal metabolism, with inverse correlation
to nutrient and sugar availability, and that the redox state of *Bc* was affected by CK both
during growth and during infection in the plant host. Our results reveal additional roles for
CK in fungus-plant interactions, and may shed light on the availability of energy and
nutrients to the fungus during initial stages of plant infection.

**Results**

**CK mediated *Bc* growth inhibition and cytoskeleton de-regulation depend on nutrient** 120 **availability**

In order to examine if direct effect of CK on *Bc* is affected by nutrient availability, we grew
*Bc* on different strengths of PDA media, with and without CK. The inhibition of mycelial
growth by CK was found to depend on the media strength, and was strongest in rich media,
slowly declining with decrease in nutrient availability (Fig. 1). CK-mediated growth inhibition
was no longer significant in 1/8 media (Fig. 1). Similar results were obtained with growth in
liquid media (Fig. S1). Growth inhibition of *Bc* by CK in rich media was previously reported by
our group (38).

To examine cytoskeleton integrity, we transformed *B. cinerea* with lifeact-GFP (39), and
proceeded to treat the transformed fungal cells with CK in full and ¼ PDB (Fig. 2). As
reported previously by our group (38), we observed mis-localization of actin, which is
normally localized to growing hyphal tips (40, 41), upon CK treatment in full PDB. However,
there was less effect of CK on F-actin distribution when cells were grown in ¼ media (Fig. 2).
Tip-specific localization of F-actin was less affected by CK in ¼ media (Fig. 2). Analysis of
corrected total fluorescence in Mock and CK treated cells grown in full and ¼ PDB
demonstrated that the ratio between actin in the tip of the cell, and the total cell, decreased
greatly in the presence of CK in full PDB but far less in ¼ PDB (Fig. 2). An important
observation was reduced actin polarization in mock samples of quarter media which might
relate to reduced growth in low strength media.

**CK mediated growth inhibition is correlated with fungal growth rate**

The observation that CK mediated growth inhibition is affected by nutrient availability (Fig.s
1-2) prompted us to examine whether this is a general phenomenon in *B. cinerea*. To do
this, we examined the correlation between growth rate and CK-mediated growth inhibition
in 17 agricultural *B. cinerea* isolates from 6 different plant hosts (Fig. 3). We found a
significant positive linear correlation between growth rate and CK mediated growth
inhibition (Fig. 3B).

**Transcriptome profiling reveals an effect of CK on *Bc* metabolic pathways and sugar** 149 **transport**

We previously conducted transcriptome profiling on *Bc* treated with CK, finding down
regulation upon CK treatment of a variety of growth and developmental pathways in *Bc*,
including inhibition of cell division, DNA replication, endocytosis and the actin cytoskeleton
(38). Given that we found the effect of CK to depend on the nutritional context (Fig. 1), we
next mined our transcriptomic data for alterations in gene expression that might explain this
phenomenon. Interestingly, we found that the glycolysis, sucrose metabolism, and oxidative
phosphorylation KEGG pathways were significantly upregulated in *Bc* upon CK treatment
((38); Fig. 4, Data S1). Over a quarter of the pathway genes were upregulated in the
glycolysis (Figure 3A) and oxidative phosphorylation (Fig. 4C) pathways, with an FDR
corrected $p\text{-val}<0.0071$ for glycolysis, and $p\text{-val}<2.94^{\text{E-11}}$ for oxidative phosphorylation. A
third of the sucrose/starch metabolism pathway was upregulated, FDR corrected $p\text{-}$
$val<0.0035$ (Fig. 4D, Data S1). Interestingly, though we found virulence genes as a group to
be downregulated upon CK treatment (38), sugar-metabolism genes known to have a role in
virulence such as pectin methyl esterase and poly-endogalacturonase (42), were
upregulated upon CK treatment, despite the overall downregulation of virulence (Fig. 4D,

Data S1). Sugar transporters are known to be upregulated in *Bc* during pathogenesis (43,
44). In addition to glycolysis, sucrose metabolism, and oxidative phosphorylation, we found
a significant upregulation of sugar transporter expression following CK treatment ((38); Data
S1).

For glycolysis pathway genes, we also conducted a RT-qPCR validation of the upregulation of
some of the key pathway genes found to be upregulated in the transcriptome following CK
treatment. Fig. 4B depicts a comparison between the fold change of these genes in the
transcriptome, and the changes we observed in an independent experiment in qPCR.

**CK rescues inhibition of glycolysis and ATP synthesis in a nutrient availability dependent** 175 **manner**

To examine the effect of CK on *Bc* metabolism, we used 2-Deoxy-D-glucose (2-DG) and
oligomycin (OM), which are inhibitors of glycolysis and ATP synthesis, respectively. 2-DG is a
glucose analog that inhibits glycolysis by competing with glucose as a substrate for
hexokinase, the rate-limiting enzyme in glycolysis. After entering the cell, 2-DG is
phosphorylated by hexokinase II to 2-deoxy-d-glucose-6-phosphate (2-DG-6-P) but, unlike
glucose, 2-DG-6-P cannot be further metabolized by phosphoglucose isomerase. This leads
to the accumulation of 2-DG-6-P in the cell, and subsequent depletion in cellular ATP (45). OM
inhibits mitochondrial H⁺-ATP-synthase, and has been attributed antifungal properties (46,
47). *Bc* was grown with these two inhibitors separately at different media strengths, as
described above. 2-DG significantly inhibited growth at low media strength, in the
concentration used (Fig. 5). OM was inhibitory at all media strengths in the concentration
used (Fig. 5). CK (100 μM) was found to rescue the inhibitory effect of 2-DG when added to
the growth media. Lowering the nutrient availability promoted CK-mediated rescue of

glycolysis inhibition by 2-DG. Interestingly, following OM treatment, the fungi became
insensitive to CK and were not further inhibited by the addition of CK. A partial rescue of
ATP synthesis mediated inhibition of growth by CK was observed in full and 1/2 media.
These results strengthen the notion that CK is affecting glycolysis in the fungus, and is in the
same pathway as ATP synthesis.

**CK induces glucose uptake in *Bc***

The dependence of CK-mediated growth inhibition on nutrition and energy state, and the
rescue of glycolysis inhibition by CK, indicated that CK is affecting *Bc* metabolism. To further
confirm this hypothesis, we next measured glucose uptake in the presence of CK in both rich
PDB and synthetic defined liquid media. We observed a significant increase of glucose
uptake in the presence of CK in both media types (Fig. 6). Interestingly, the percent increase
of glucose uptake by *Bc* in the presence of CK increased with decreasing nutrient availability
in the media (Fig. 6). There was significant increase of sugar uptake in defined media but it
was not dependent on media strength. We know 2-DG competes with glucose in the
glycolytic pathway. The increase of sugar uptake might be the reason why CK rescues
glycolysis inhibition by 2-DG. Since ATP is required for growth, it is not surprising that
inhibition of ATP synthesis prevented further growth inhibition by CK.

**CK alters *Bc* redox status**

Since metabolic pathway fluctuations can affect redox status (48), and redox homeostasis in
*Bc* is known to change during host infection (49, 50), we next examined cytosolic and
mitochondrial redox status in the fungus grown with CK. For this purpose, we generated *Bcl*-
16 strain lines expressing GRX-roGFP and mito-roGFP, using previously described expression

cassettes that were used for the measurement of *Bc* redox status (39, 51). We found that
after 24 hours of growth, CK significantly altered the cytosolic redox of *Bc* to a more reduced
state, while the mitochondrial redox was significantly more oxidised with CK (Fig. 7A).
Following redox over time, we observed similar states in the first 8 hours of growth, with CK
starting to affect the redox state after about 15 hours of co-cultivation, corresponding with
the stage at which mycelia are elongating (Fig. 7B-C). Cytosolic and mitochondrial redox
states are often inversely correlated (52). Interestingly, there was an inverse effect of CK on
cytosolic and mitochondrial redox of growing mycelia (Fig. 7).

**Endogenous CK content of tomato leaves affects redox state of *Bc* during plant infection**

Since CK affected redox in *Bc* in rich media, we next examined whether endogenous CK
content in tomato leaves can affect the redox status of infecting *Bc* mycelia. For this, *Bc*
GRX-roGFP and mito-roGFP conidia from freshly sporulated PDA plates were used to infect
detached leaves from M82, IPT, and CKX plants. Leaves overexpressing IPT have increased
CK content and are more resistant to *Bc* infection, while leaves overexpressing CKX have
reduced CK content and are more sensitive to *Bc* infection (4). Redox-dependent changes in
GRX-roGFP2 and mito-roGFP fluorescence in living *Botrytis* hyphae have been previously
visualized by confocal laser scanning microscopy (CLSM) (52). Infecting *Bc* hyphae expressing
GRX-roGFP in the cytosol or mito-roGFP in the intermembrane mitochondrial space were
analysed microscopically, 24 and 48 hours after inoculation. Similar to the fluorometry-
based calculations, a higher 405nm/488nm ratio indicates more oxidised state, and a lower
ratio, a more reduced state. We found that 24 hours after inoculation, the cytosolic redox
state of the infecting hyphae was more oxidised on IPT leaves (high CK content) as
compared to the infecting hyphae on mock M82 leaves, while infecting hyphae on CKX (low

CK content) were more reduced (Fig. 8A,C). In a parallel set of experiments which included
additional genotypes, we also observed increased oxidation of the *Bc* cytosol when infecting
M82 leaves that were pre-treated with CK, or when infecting leaves of the hypersensitive
*clausa* mutant (Fig. S2). 48 hours post inoculation, we observed an opposite trend of the
cytosolic redox of the infecting hyphae, with hyphae on IPT becoming more reduced while
hyphae on CKX were more oxidised, when compared with M82 leaves. (Fig. 8A,C). Here,
again, in a parallel set of experiments, which included additional genotypes, we also
observed increased reduction of the *Bc* cytosol when infecting leaves of the hypersensitive
*clausa* mutant (Fig. S2). When examining mitochondrial redox of the hyphae growing on IPT
or CKX leaves, we found that, 48 post inoculation, mitochondrial redox of the hyphae
growing on IPT was significantly oxidised as compared to mock M82, while hyphae growing
on CKX were significantly reduced (Fig. 8B,C). The cytosolic redox was measured in a
parallel set of experiments on leaf discs co-cultivated with *Bc* spores in a fluorimeter plate,
with similar results (Fig. S3). To verify that the virulence of the roGFP fungi was intact, we
also conducted disease assays of these fungi on the different genotypes, with findings
consistent with previous results (4), i.e., reduced disease on high-CK or CK-hypersensitive
genotypes, and increased disease on low-CK genotypes (Fig. S4).

We further examined the transcriptome of *Bc* grown in the presence of tobacco seedlings
following CK treatment, finding significant changes, both significant downregulation and
significant upregulation, in NADPH/NADH reductases and oxidoreductases (Fig. S5). These
transcriptional changes in the fungus further support the notion that CK is affecting ROS
coping mechanisms in *Bc*, and could underlie the altered pathogenesis courses observed in
tomato genotypes with altered CK content.

Discussion

It has been previously reported by us and others that CK promotes fungal disease
resistance in plants (4, 12). Recently, we reported a direct inhibitory effect of CK on *Bc*
growth and development *in vitro* (38). Our previous results indicated that *B. cinerea*
responds to CK and activates signaling cascades in its presence, leading to inhibition of the
cell cycle, mis-localization of the actin cytoskeleton, and inhibition of cellular trafficking (38).
We hypothesized that CK may be exerting its effect through influence on fungal metabolic
pathways. The present study was performed to examine the effect of CK on *Bc* metabolism.
We examined sugar uptake, glycolysis, and cellular redox status of *Bc* in the presence of CK.
Our results demonstrate that the inhibitory activity of CK against *Bc* is largely dependent on
the status of nutrient and energy availability.

*Inhibition by CK is dependent on fungal nutrition status*

We found that the inhibitory effect of CK on *Bc* was correlated with nutrient and
energy availability, with fungi grown in rich media being inhibited more strongly than fungi
grown in sub-optimal conditions (Fig. 1, S1). Intact F-actin was found to be required for
hyphal growth, morphogenesis, and virulence, which were all impaired in F-actin capping
protein deletion mutants (53). Since we had previously observed that CK caused mis-
localization of actin at the growing tip of *Bc* hyphae (38), we examined whether this
phenomenon was also correlated with the nutritional status of the environment. Indeed, we
found that the effect of CK on the cytoskeleton is also dependent on the status of nutrient
availability (Fig. 2). Interestingly, we observed reduced polarization of F-actin in *Bc* growing
in minimal media when compared with rich media (Fig. 2), providing one underlying
mechanism for reduced fungal growth under low nutrient conditions.

*CK promotes fungal metabolism*

Our previously published transcriptome profile revealed that important
developmental pathways such as cell division, cellular trafficking, and the cytoskeleton,
were inhibited upon CK treatment. Our results demonstrated that the effect of CK on *Bc* is
dependent on nutritional status. Re-examining our transcriptomic data in light of this, we
found that glycolysis, sucrose metabolism and oxidative phosphorylation pathways were
significantly enriched upon CK treatment (Fig. 4). The expression of sugar transporters was
also significantly upregulated.

This transcriptomic data was generated under defined nutrient conditions. To
further examine possible effects of CK on fungal metabolism, we investigated the effect of
CK on *Bc* glycolysis and ATP synthesis, under different nutrient and energy availability. We
found that CK was able to rescue inhibition of glycolysis and ATP synthesis in a nutrient
dependent manner, with stronger rescue observed under minimal nutrient conditions. CK
also promoted an increase in sugar uptake by the fungus, in a nutrient dependent manner,
with the strongest uptake promotion observed under minimal nutrient and energy
conditions (Figs. 4-5). Upregulation of glycolysis and oxidative phosphorylation key genes,
together with the increased uptake of sugar, could explain the rescue of metabolic
inhibition by CK. Taken together, these results confirm that the effect of CK on *Bc* is
dependent on nutrient availability.

*CK alters fungal redox*

Changes in metabolic pathways are often reflected in the redox status (50). Hence,
we examined the effect of CK on *Bc* redox status both *in vitro* and *in planta*, using tomato
genotypes with varying CK content or sensitivity. *Bc* cells grown with CK in rich media had a
significantly reduced cytosol and a significantly oxidized mitochondria (Fig. 7). A reduced
cytosol and oxidized mitochondria is indicative of increased glycolysis and oxidative

phosphorylation (54, 55). Thus, this result correlated with our transcriptomic data, in which
glycolysis and oxidative phosphorylation are upregulated in the presence of CK (Fig. 4), and
also with our results demonstrating that CK promotes sugar uptake in *Bc* (Fig. 6). *In planta*,
after 48h of inoculation, *Bc* had a reduced cytosol and oxidized mitochondria when infecting
the CK-rich IPT, and an oxidized cytosol and reduced mitochondria when infecting the CK-
deficient CKX, confirming that CK can affect the redox state of *Bc* during infection *in planta*.
This significant change in redox was also coupled with the lower virulence on IPT leaves and
higher virulence on CKX leaves (Fig. S4, (4)). It was previously reported that the cytosol of
infecting *Bc* hyphae on dead onion peel is reduced (52). Our results also show that *Bc*
cytosol is more reduced in IPT, but the resultant infection was lower when compared to that
on CKX. In addition to the different hosts systems and different infection time frames, a
possible explanation for this could be the CK-mediated immunity induced in the host (4, 12).
The results of redox state of *Bc* hyphae in vitro and in planta, together with transcriptome
data and sugar uptake results, might relate to the role of CK in nutrient allocation in fungi
during infection.

*CK defines and supports tissues for fungal use*

CKs have previously described roles in plant–pathogen interactions (3, 4). The
interaction of some biotrophic pathogens with their hosts leads to the formation of green
bionissia (formerly known as green islands) (56), which are sites of green living tissue
surrounding the sites of active pathogen growth (9). The formation of these green bionissia
is correlated with elevated levels of cytokinins in these tissues. It is believed that cytokinins
likely delay the onset of senescence in green bionissia, allowing pathogens access to more
nutrients from the plant (9). Necrotrophic fungal pathogens, which obtain their nutrients
from dead plant cells, have also been reported to cause the formation of green tissue

around the sites of infection (green necronissia) in certain cases (9). Application of
exogenous CK is known to induce the formation green necronissia (9, 15, 57). We found
that the effect of CK on *Bc* is dependent on nutrient availability, and can induce glycolysis,
oxidative phosphorylation and sugar uptake. We also observed CK-mediated redox shifts in
*Bc*, that are likely due to these increases in metabolism.

[revised manuscript text omitted]

**Mycelial growth assay**

To study how nutrient availability affects mycelial growth of *Bc* in the presence of CK, 6-BAP
(6-Benzylaminopurine, Sigma-Aldrich) was dissolved in 10mM NaOH and added to PDA
media of full, half, one-fourth and one-eighth strength. To study how *Bc* responds to
metabolic inhibitors in the presence of CK, 2-Deoxy-D-glucose (2-DG) and oligomycin (OM)
were added to PDA media at above mentioned strength to final concentration 100 μ M, 2.5
mM and 0.1 μ g/mL respectively. *Bc* mycelial plugs (5 mm) taken ~1cm from the edge of a
fresh plate were placed at the centre of PDA plates and incubated under the above
mentioned growth conditions. To measure the mycelium weight in liquid media, *Bc* was
cultured in stationary liquid PDB full and one-fourth media strength in the presence of 100
μ M concentrations of 6-BAP. After 72 h, the fungal mass was dried and the dry weight was
measured.

**Effect of CK on glucose uptake at different media strength**

To evaluate the effect of CK on glucose uptake, spores were harvested in 1 mg mL⁻¹ glucose
and 1 mg mL⁻¹ K₂HPO₄ and filtered through 40 µm pore cell strainer. Spore concentration
was adjusted to 10⁶ spores mL⁻¹ using a Neubauer chamber. *Bc* spores were grown in potato
dextrose broth (PDB) or defined media of full, half and one-fourth strength. Composition of
defined media was glucose (20 g/L) and 4 g/L each of K₂HPO₄, KH₂PO₄, and NH₄SO₄. 100 µM
of CK were added to both PDB and defined media cultures, which were then allowed to
grow for 48 hours. The amount of metabolized glucose was analysed by measuring the
amount of glucose present in the media by standard DNSA method (65) using dextrose as
standard. Metabolized glucose was assumed to be inversely proportional to the amount
present in the media. For control, glucose in different strength of media subjected to above
mentioned conditions but without *Bc* was measured.

**Generation of *B. cinerea* lines expressing lifeact-GFP**

[revised manuscript text omitted]

520 then returned to the dark for sporulation. Spores were harvested in 1 mg mL^{-1} glucose and 1
521 mg mL^{-1} K_2HPO_4 , and filtered through $40 \mu\text{m}$ pore cell strainer. Spore concentration was
522 adjusted to 10^6 spores mL^{-1} after quantification under a light microscope using a Neubauer
chamber. Leaflets 10-15 days old tomato plants were excised and immediately placed in
humid chambers. Leaflets were inoculated with one droplets of $5 \mu\text{L}$ suspension. Twenty-
four hours after inoculation, germinated conidia were imaged using a fluorescent Olympus
IX 81 inverted laser scanning confocal microscope (Fluoview 500). Images were collected
with a $60\times$ 1.0 NA PlanApo water-immersion lens in multi-track line mode. roGFP was
excited at 405 nm in the first track and at 488 nm in the second track. For both excitation
wavelengths, roGFP fluorescence was collected with a bandpass filter of 505–530 nm and
averaged from four readings for noise reduction (52). The Ratiometric analyses of
fluorescence images were calculated using Fiji-ImageJ.

**Data analysis**

Data is presented as minimum to maximum values in boxplots or floating bar graphs, or as
average \pm SEM in bar graphs. For Gaussian distributed samples, we analyzed the statistical
significance of differences between two groups using a two-tailed t-test, with additional
post hoc correction where appropriate, such as Welch's correction for t-tests between
samples with unequal variances. We analyzed the statistical significance of differences
among three or more groups using analysis of variance (ANOVA). Regular ANOVA was used
for groups with equal variances, and Welch's ANOVA for groups with unequal variances.
Significance in differences between the means of different samples in a group of three or
more samples was assessed using a post-hoc test. The Tukey post-hoc test was used for
samples with equal variances, when the mean of each sample was compared to the mean of
every other sample. The Bonferroni post-hoc test was used for samples with equal
variances, when the mean of each sample was compared to the mean of a control sample.
The Dunnett post-hoc test was used for samples with unequal variances. For samples with
non-Gaussian distribution, we analyzed the statistical significance of differences between
two groups using a Mann-Whitney U test, and the statistical significance of differences
among three or more groups using Kruskal-Wallis ANOVA, with Dunn's multiple comparison
post-hoc test as indicated. Gaussian distribution or lack thereof was determined using the
Shapiro-Wilk test for normality. Statistical analyses were conducted using Prism8™.

**Acknowledgments**

This work was supported by the Israel Science Foundation grant No. 1759/20 to MB. The
funder had no role in study design, data collection and analysis, decision to publish, or

preparation of the manuscript. The lifeact and roGFP constructs were kindly provided by
Julia Schumacher. GA is supported by the Indo-China ARO Postdoctoral Fellowship Program.
MB thanks members of the Bar group for continuous discussion and support.

**Author contributions**

Conceptualization: GA, MB. Design: GA, RG, MB. Methodology & experimentation: GA, RG.
Analysis: GA, RG, MB. Manuscript: GA, MB.

**The authors declare no competing interest.**

**Data availability statement**

The authors declare that the data supporting the findings of this study are available within
the paper and its supplementary information files. Raw data is available from the
corresponding author upon reasonable request. The raw data generated in the
transcriptomic analyses is deposited in NCBI under Bioproject accession number
PRJNA718329.

**Supplementary information**

**Fig. S1.** CK-mediated growth inhibition depends on nutrient availability- Dry weight
measured from fungi grown in liquid media.

**Fig. S2.** Plant endogenous CK alters *Bc* cytosolic redox state during infection- additional
genotypes.

**Fig. S3.** Plant endogenous CK alters *Bc* redox state during infection- plate assay.

**Fig. S4.** *Bc* transformed roGFP lines display virulence behaviour similar to that of the
background line.

**Fig. S5.** CK alters *Bc* redox state- changes in the transcriptome.

**Table S1.** Oligonucleotides used for generating and validating *Botrytis cinerea* transgenic
strains.

**Table S2.** Primers used in RT-qPCR.

**Data S1.** Transcriptomic effect of CK on *Bc* metabolic pathways.

**References**

- 1. Wybouw B, de Rybel B. 2019. Cytokinin – A Developing Story. Trends in Plant Science
<https://doi.org/10.1016/j.tplants.2018.10.012>.
- 2. Cortleven A, Leuendorf JE, Frank M, Pezzetta D, Bolt S, Schmölling T. 2019. Cytokinin action in
response to abiotic and biotic stresses in plants. Plant Cell and Environment
<https://doi.org/10.1111/pce.13494>.
- 3. Akhtar SS, Mekureyaw MF, Pandey C, Roitsch T. 2020. Role of Cytokinins for Interactions of
Plants With Microbial Pathogens and Pest Insects. Frontiers in Plant Science
<https://doi.org/10.3389/fpls.2019.01777>.
- 4. Gupta R, Pizarro L, Leibman-Markus M, Marash I, Bar M. 2020. Cytokinin response induces
immunity and fungal pathogen resistance, and modulates trafficking of the PRR LeEIX2 in
tomato. Molecular Plant Pathology 21:1287–1306.
- 5. Brenner WG, Romanov GA, Köllmer I, Bürkle L, Schmölling T. 2005. Immediate-early and
delayed cytokinin response genes of *Arabidopsis thaliana* identified by genome-wide
expression profiling reveal novel cytokinin-sensitive processes and suggest cytokinin action
through transcriptional cascades. Plant Journal 44:314–333.
- 6. Sakakibara H, Takei K, Hirose N. 2006. Interactions between nitrogen and cytokinin in the
regulation of metabolism and development. Trends in Plant Science
<https://doi.org/10.1016/j.tplants.2006.07.004>.
- 7. Martín AC, del Pozo JC, Iglesias J, Rubio V, Solano R, de La Peña A, Leyva A, Paz-Ares J. 2000.
Influence of cytokinins on the expression of phosphate starvation responsive genes in
*Arabidopsis*. Plant Journal 24:559–567.

- 8. Maruyama-Nakashita A, Nakamura Y, Yamaya T, Takahashi H. 2004. A novel regulatory
pathway of sulfate uptake in Arabidopsis roots: Implication of CRE1/WOL/AHK4-mediated
cytokinin-dependent regulation. *Plant Journal* 38:779–789.
- 9. Walters DR, McRoberts N, Fitt BDL. 2008. Are green islands red herrings? Significance of
green islands in plant interactions with pathogens and pests. *Biological Reviews*
<https://doi.org/10.1111/j.1469-185X.2007.00033.x>.
- 10. Babosha A v. 2009. Regulation of resistance and susceptibility in wheat-powdery mildew
pathosystem with exogenous cytokinins. *Journal of Plant Physiology* 166:1892–1903.
- 11. Sharma N, Rahman MH, Liang Y, Kav NNV. 2010. Cytokinin inhibits the growth of
*Leptosphaeria maculans* and *Alternaria brassicae*. *Canadian Journal of Plant Pathology*
32:306–314.
- 12. Choi J, Huh SU, Kojima M, Sakakibara H, Paek KH, Hwang I. 2010. The cytokinin-activated
transcription factor ARR2 promotes plant immunity via TGA3/NPR1-dependent salicylic acid
signaling in arabidopsis. *Developmental Cell* 19:284–295.
- 13. Jameson PE. 2000. Cytokinins and auxins in plant-pathogen interactions - An overview, p.
369–380. *In* *Plant Growth Regulation*.
- 14. Greene EM. 1980. Cytokinin production by microorganisms. *The Botanical Review* 46:25–74.
- 15. Angra R, Mandahar CL. 1991. Pathogenesis of barley leaves by *Helminthosporium teres* I:
Green island formation and the possible involvement of cytokinins. *Mycopathologia* 114:21–
27.
- 16. Liu Z, Bushnell WR. 1986. Effects of cytokinins on fungus development and host response in
powdery mildew of barley. *Physiological and Molecular Plant Pathology* 29:41–52.
- 17. Ashby AM. 2000. Biotrophy and the cytokinin conundrum. *Physiological and Molecular Plant*
*Pathology* <https://doi.org/10.1006/pmpp.2000.0294>.
- 18. Milo GE, Srivastava BIS. 1969. Effect of cytokinins on tobacco mosaic virus production in local-
lesion and systemic hosts. *Virology* 38:26–31.
- 19. Großkinsky DK, Naseem M, Abdelmohsen UR, Plickert N, Engelke T, Griebel T, Zeier J, Novák
O, Strnad M, Pfeifhofer H, van der Graaff E, Simon U, Roitsch T. 2011. Cytokinins mediate
resistance against *Pseudomonas syringae* in tobacco through increased antimicrobial
phytoalexin synthesis independent of salicylic acid signaling. *Plant Physiology* 157:815–830.
- 20. Ballaré CL. 2011. Jasmonate-induced defenses: A tale of intelligence, collaborators and
rascals. *Trends in Plant Science* <https://doi.org/10.1016/j.tplants.2010.12.001>.
- 21. Gupta R, Leibman-Markus M, Pizarro L, Bar M. 2021. Cytokinin induces bacterial pathogen
resistance in tomato. *Plant Pathology* 70:318–325.

- 22. Fillinger S, Elad Y. 2015. Botrytis - The fungus, the pathogen and its management in
agricultural systems Botrytis - The Fungus, the Pathogen and its Management in Agricultural
Systems.
- 23. Veloso J, van Kan JAL. 2018. Many Shades of Grey in Botrytis–Host Plant Interactions. Trends in
Plant Science <https://doi.org/10.1016/j.tplants.2018.03.016>.
- 24. Rajarammohan S. 2021. Redefining Plant-Necrotroph Interactions: The Thin Line Between
Hemibiotrophs and Necrotrophs. *Frontiers in Microbiology* 12:944.
- 25. Dean R, van Kan JAL, Pretorius ZA, Hammond-Kosack KE, di Pietro A, Spanu PD, Rudd JJ,
Dickman M, Kahmann R, Ellis J, Foster GD. 2012. The Top 10 fungal pathogens in molecular
plant pathology. *Molecular Plant Pathology* [https://doi.org/10.1111/j.1364-](https://doi.org/10.1111/j.1364-3703.2011.00783.x)
3703.2011.00783.x.
- 26. Colmenares AJ, Aleu J, Durán-Patrón R, Collado IG, Hernández-Galán R. 2002. The putative
role of botrydial and related metabolites in the infection mechanism of *Botrytis cinerea*.
*Journal of Chemical Ecology* 28:997–1005.
- 27. Tani H, Koshino H, Sakuno E, Nakajima H. 2005. Botcinins A, B, C, and D, Metabolites
Produced by *Botrytis cinerea*, and Their Antifungal Activity against *Magnaporthe oryzae*, a
Pathogen of Rice Blast Disease. *Journal of Natural Products* 68.
- 28. Choquer M, Fournier E, Kunz C, Levis C, Pradier JM, Simon A, Viaud M. 2007. *Botrytis cinerea*
virulence factors: New insights into a necrotrophic and polyphageous pathogen. *FEMS*
*Microbiology Letters* <https://doi.org/10.1111/j.1574-6968.2007.00930.x>.
- 29. Williamson B, Tudzynski B, Tudzynski P, van Kan JAL. 2007. *Botrytis cinerea*: The cause of grey
mould disease. *Molecular Plant Pathology* [https://doi.org/10.1111/j.1364-](https://doi.org/10.1111/j.1364-3703.2007.00417.x)
3703.2007.00417.x.
- 30. van Kan JAL. 2006. Licensed to kill: the lifestyle of a necrotrophic plant pathogen. *Trends in*
*Plant Science* <https://doi.org/10.1016/j.tplants.2006.03.005>.
- 31. Schumacher J, Simon A, Cohrs KC, Viaud M, Tudzynski P. 2014. The Transcription Factor
BcLTF1 Regulates Virulence and Light Responses in the Necrotrophic Plant Pathogen *Botrytis*
*cinerea*. *PLoS Genetics* 10:e1004040.
- 32. T. Kosuge, W.B. Hewitt. 1964. Exudates of grape berries and their effect on germination of
conidia of *Botrytis cinerea*. *Phytopathology* 54:167–172.
- 33. R.G. Orellana, C.A. Thomas. 1962. Nature of predisposition of castor beans to *Botrytis*. I.
Relation of leachable sugar and certain other biochemical constituents of the capsule to
varietal susceptibility. *Phytopathology* 52:533–538.
- 34. Solomon PS, Tan KC, Oliver RP. 2003. The nutrient supply of pathogenic fungi; a fertile field
for study. *Molecular Plant Pathology* <https://doi.org/10.1046/j.1364-3703.2003.00161.x>.

- 35. Kimura A, Takano Y, Furusawa I, Okuno T. 2001. Peroxisomal Metabolic Function Is Required
for Appressorium-Mediated Plant Infection by *Colletotrichum lagenarium*. *The Plant Cell*
13:1945–1957.
- 36. Joosten MHAJ, Hendrickx LJM, de Wit PJGM. 1990. Carbohydrate composition of apoplastic
fluids isolated from tomato leaves inoculated with virulent or avirulent races of *Cladosporium*
*fulvum* (syn. *Fulvia fulva*). *Netherlands Journal of Plant Pathology* 96:103–112.
- 37. Bowyer P, Mueller E, Lucas J. 2000. Use of an isocitrate lyase promoter-GFP fusion to monitor
carbon metabolism of the plant pathogen *Tapesia yallundae* during infection of wheat.
*Molecular Plant Pathology* 1:253–262.
- 38. Gupta R, Anand G, Pizarro L, Laor D, Kovetz N, Sela N, Yehuda T, Gazit E, Bar M. 2021.
Cytokinin Inhibits Fungal Development and Virulence by Targeting the Cytoskeleton and
Cellular Trafficking. *mBio* <https://doi.org/10.1128/mbio.03068-20>.
- 39. Schumacher J. 2012. Tools for *Botrytis cinerea*: New expression vectors make the gray mold
fungus more accessible to cell biology approaches. *Fungal Genetics and Biology* 49:483–497.
- 40. Bepewiki A, Lichius A, Read ND. 2011. Actin organization and dynamics in filamentous fungi.
*Nature Reviews Microbiology* <https://doi.org/10.1038/nrmicro2666>.
- 41. Walker SK, Garrill A. 2006. Actin microfilaments in fungi. *Mycologist* 20:26–31.
- 42. ten Have A, Mulder W, Visser J, van Kan JAL. 1998. The endopolygalacturonase gene *Bcpg1* is
required to full virulence of *Botrytis cinerea*. *Molecular Plant-Microbe Interactions* 11:1009–
1016.
- 43. Dulermo T, Rasclé C, Chinnici G, Gout E, Bligny R, Cotton P. 2009. Dynamic carbon transfer
during pathogenesis of sunflower by the necrotrophic fungus *Botrytis cinerea*: From plant
hexoses to mannitol. *New Phytologist* 183:1149–1162.
- 44. Rui O, Hahn M. 2007. The *Botrytis cinerea* hexokinase, *Hxk1*, but not the glucokinase, *Glk1*, is
required for normal growth and sugar metabolism, and for pathogenicity on fruits.
*Microbiology* 153:2791–2802.
- 45. Pajak B, Siwiak E, Sołtyka M, Priebe A, Zieliński R, Fokt I, Ziemniak M, Jaśkiewicz A, Borowski
R, Domoradzki T, Priebe W. 2020. 2-Deoxy-D-Glucose and its analogs: From diagnostic to
therapeutic agents. *International Journal of Molecular Sciences*
<https://doi.org/10.3390/ijms21010234>.
- 46. Shchepina LA, Pletjushkina OY, Avetisyan A v, Bakeeva LE, Fetisova EK, Izyumov DS,
Saprunova VB, Vyssokikh MY, Chernyak B v, Skulachev VP. 2002. Oligomycin, inhibitor of the
F₀ part of H⁺-ATP-synthase, suppresses the TNF-induced apoptosis. *Oncogene* 21.
- 47. SMITH RM, PETERSON WH, McCOY E. 1954. Oligomycin, a new antifungal antibiotic.
*Antibiotics & chemotherapy* 4:962–970.

- 48. Ralser M, Wamelink MM, Kowald A, Gerisch B, Heeren G, Struys EA, Klipp E, Jakobs C,
Breitenbach M, Lehrach H, Krobitsch S. 2007. Dynamic rerouting of the carbohydrate flux is
key to counteracting oxidative stress. *Journal of Biology* 6:10.
- 49. Li H, Tian S, Qin G. 2019. NADPH oxidase Is crucial for the cellular redox homeostasis in fungal
pathogen botrytis cinerea. *Molecular Plant-Microbe Interactions* 32:1508–1516.
- 50. Viefhues A, Heller J, Temme N, Tudzynski P. 2014. Redox systems in Botrytis cinerea: Impact
on development and virulence. *Molecular Plant-Microbe Interactions* 27:858–874.
- 51. Marschall R, Schumacher J, Siegmund U, Tudzynski P. 2016. Chasing stress signals - Exposure
to extracellular stimuli differentially affects the redox state of cell compartments in the wild
type and signaling mutants of Botrytis cinerea. *Fungal Genetics and Biology* 90:12–22.
- 52. Heller J, Meyer AJ, Tudzynski P. 2012. Redox-sensitive GFP2: Use of the genetically encoded
biosensor of the redox status in the filamentous fungus Botrytis cinerea. *Molecular Plant
Pathology* 13:935–947.
- 53. González-Rodríguez VE, Garrido C, Cantoral JM, Schumacher J. 2016. The F-actin capping
protein is required for hyphal growth and full virulence but is dispensable for septum
formation in Botrytis cinerea. *Fungal Biology* 120:1225–1235.
- 54. Li H, Zhang Z, He C, Qin G, Tian S. 2016. Comparative proteomics reveals the potential targets
of BcNoxR, a putative regulatory subunit of NADPH oxidase of botrytis cinerea. *Molecular
Plant-Microbe Interactions* 29:990–1003.
- 55. Muller FL, Liu Y, van Remmen H. 2004. Complex III releases superoxide to both sides of the
inner mitochondrial membrane. *Journal of Biological Chemistry* 279:49064–49073.
- 56. Coghlan SE, Walters DR. 1992. Photosynthesis in green-islands on powdery mildewinfected
barley leaves. *Physiological and Molecular Plant Pathology* 40:31–38.
- 57. Angra-Sharma R, Sharma DK. 2000. Cytokinins in pathogenesis and disease resistance of
Pyrenophora teres-barley and Dreschlera maydis-maize interactions during early stages of
infection. *Mycopathologia* 148:87–95.
- 58. Prins TW, Tudzynski P, von Tiedemann A, Tudzynski B, ten Have A, Hansen ME, Tenberge K,
van Kan JAL. 2000. Infection Strategies of Botrytis cinerea and Related Necrotrophic
Pathogens, p. 33–64. *In Fungal Pathology*.
- 59. Barnes SE, Shaw MW. 2002. Factors affecting symptom production by latent Botrytis cinerea
in Primulaxpolyantha. *Plant Pathology* 51:746–754.
- 60. Vitorino Borges Á, Moreira Saraiva R, Antonio Maffia L. 2014. Key factors to inoculate Botrytis
cinerea in tomato plants. *Summa Phytopathologica* 221–225.
- 61. Dik AJ, Wubben JP. 2007. Epidemiology of Botrytis cinerea Diseases in Greenhouses. *Botrytis:
Biology, Pathology and Control* 319–333.

- 62. Sirjusingh C, Sutton JC, Tsujita MJ. 1996. Effects of inoculum concentration and host age on
infection of geranium by *Botrytis cinerea*. *Plant Disease* 80:154–159.
- 63. Elad Y, Williamson B, Tudzynski P, Delen N. 2007. *Botrytis: Biology, pathology and control*.
*Botrytis: Biology, Pathology and Control* 1–403.
- 64. Li B, Wang R, Wang S, Zhang J, Chang L. 2021. Diversified Regulation of Cytokinin Levels and
Signaling During *Botrytis cinerea* Infection in *Arabidopsis*. *Frontiers in Plant Science* 12.
- 65. Miller GL. 1959. Use of Dinitrosalicylic Acid Reagent for Determination of Reducing Sugar.
*Analytical Chemistry* 31.
- 66. Gupta R, Anand G, Pizarro L, Laor D, Kovetz N, Sela N, Yehuda T, Gazit E, Bar M. 2021.
Cytokinin inhibits fungal development and virulence by targeting the cytoskeleton and
cellular trafficking. *bioRxiv*, <https://doi.org/10.1101/20201104369215>.
- 67. Levis C, Fortini D, Brygoo Y. 1997. Transformation of *Botrytis cinerea* with the nitrate
reductase gene (*niaD*) shows a high frequency of homologous recombination. *Current*
*Genetics* 32:157–162.
- 68. Schindelin J, Arganda-Carreras I, Frise E, Kaynig V, Longair M, Pietzsch T, Preibisch S, Rueden
C, Saalfeld S, Schmid B, Tinevez J-Y, White DJ, Hartenstein V, Eliceiri K, Tomancak P, Cardona
760 A. 2012. Fiji: an open-source platform for biological-image analysis. *Nature Methods* 9:676–
761 682.
- 69. Love MI, Huber W, Anders S. 2014. Moderated estimation of fold change and dispersion for
RNA-seq data with DESeq2. *Genome Biology* 15:550.
- 70. Buchfink B, Xie C, Huson DH. 2014. Fast and sensitive protein alignment using DIAMOND.
*Nature Methods* <https://doi.org/10.1038/nmeth.3176>.
- 71. Conesa A, Götz S, García-Gómez JM, Terol J, Talón M, Robles M. 2005. Blast2GO: A universal
tool for annotation, visualization and analysis in functional genomics research. *Bioinformatics*
21:3674–3676.
- 72. Xie C, Mao X, Huang J, Ding Y, Wu J, Dong S, Kong L, Gao G, Li CY, Wei L. 2011. KOBAS 2.0: A
web server for annotation and identification of enriched pathways and diseases. *Nucleic*
*Acids Research* 39:316–322.
- 73. Silva-Moreno E, Brito-Echeverría J, López M, Ríos J, Balic I, Campos-Vargas R, Polanco R. 2016.
Effect of cuticular waxes compounds from table grapes on growth, germination and gene
expression in *Botrytis cinerea*. *World Journal of Microbiology and Biotechnology* 32:74.
- 74. Llanos A, François JM, Parrou JL. 2015. Tracking the best reference genes for RT-qPCR data
normalization in filamentous fungi. *BMC Genomics* 16:71.
- 75. D'haene B, Vandesompele J, Hellemans J. 2010. Accurate and objective copy number profiling
using real-time quantitative PCR. *Methods* <https://doi.org/10.1016/j.ymeth.2009.12.007>.

**Figure legends**781 **Fig. 1 CK-mediated growth inhibition depends on nutrient availability**

*Bc* mycelia were grown on PDA plates without (Mock) or with the addition of the CK 6-BAP
(6-Benzylaminopurine, 100 μ M) and incubated at 22 ± 2 °C in the dark. Mycelial area was
measured after 5 days. Boxplots are shown with minimum to maximum values, inner
quartile ranges (box), median (line in box), and outer quartile ranges (whiskers), N=6.
Results were analyzed for statistical significance using a one-way ANOVA with a Bonferroni
post-hoc test, or a two-tailed t-test with Welch's correction. Asterisks indicate statistically
significant differences between the Mock and CK samples within the same media,
**** p <0.0001; * p <0.05; ns=non-significant. Upper case letters indicate statistically
significant differences in the growth of Mock samples in different media, p <0.0035; lower
case letters indicate statistically significant differences in the growth of CK-treated samples
in different media, p <0.05.

**Fig. 2 CK-mediated cytoskeleton inhibition depends on nutrient availability**

Spores of *B. cinerea* expressing the filamentous actin marker lifeact-GFP, were treated with
Mock or CK, and grown for 6 h and 24h hours prior to confocal visualization, in full and one
forth potato dextrose broth media, respectively. **(A)** Representative images, bar=10 μ M. **(B)**
Analysis of corrected total fluorescence (CTF) of the ratio between actin at the tip of the cell
and the total cell in Mock and CK treated cells. Three independent experiments were
conducted with a minimum of 24 images analyzed, N>32 growing hypha tips. Bars are
shown \pm SEM, with all points. Letters and Asterisks indicate significance in Kruskal-Wallis
ANOVA with Dunn's post hoc test, * p <0.05 and **** p <0.0001.

**Fig. 3 CK-mediated growth inhibition levels depend on growth rate**

*Bc* mycelia from different isolates were grown on PDA plates without (Mock) or with the
addition of the CK 6-BAP (6-Benzylaminopurine, 100 μ M) and incubated at 22 ± 2 °C in the
dark. Mycelial area was measured after 5 days. Experiment was repeated 3 independent
806 times, N=9.

**A** growth rate per day of several isolates with or without CK. Bars depict Mean \pm SEM, all
points shown. Results were analyzed for statistical significance using a one-way ANOVA with
a Bonferroni post-hoc test. Asterisks indicate statistically significant differences between the
Mock and CK samples of each isolate, **** p <0.0001. The average % of CK mediated
inhibition is indicated above the CK bar for each isolate.

**B** Regression analysis of the relationship between growth rate and CK-mediated inhibition in
17 different *B. cinerea* isolates from tomato, pepper, eggplant, cucumber, grapevine, and
strawberry. Simple linear regression among samples was found significantly different from
zero, $p=0.0052$.

**Fig. 4 Transcriptomic analysis of botrytis grown with CK reveals up-regulation of energy**
**metabolism pathways.**

Illumina Hiseq NGS was conducted on *Bc* Mock treated or CK treated samples, 3 biological
repeats each. Gene expression values were computed as FPKM, and differential expression
analysis was completed using the DESeq2 R package. Genes with an adjusted p -value of no
more than 0.05 and \log_2FC (Fold Change) greater than 1 or lesser than -1 were considered
differentially expressed. The KOBAS 3.0 tool was used to detect the statistical enrichment of
differential expression genes in Kyoto Encyclopedia of Genes and Genomes (KEGG)
pathways and Gene Ontology (GO). Pathways were tested for significant enrichment using
Fisher's exact test, with Benjamini and Hochberg FDR correction. Corrected p -value was
deemed significant at $p<0.05$. The Glycolysis (**A-B**), Oxidative phosphorylation (**C**) and
Sucrose metabolism (**D**) pathways were all found to be significantly up-regulated upon CK
treatment. See also Supplemental data 1. **A,C,D** Heatmap representation of upregulated
genes in the CK transcriptome in each indicated pathway. **B** Comparison of RT-qPCR
validation of the 4 indicated key glycolysis genes with the transcriptomic values. The full
transcriptome data was previously published (Gupta et al., 2021) and is available (NCBI
bioproject PRJNA718329).

**Fig. 5 CK rescues glycolysis inhibition and partially rescues ATP synthesis inhibition**

*Bc* mycelia were grown on PDA plates without (Mock) or with the addition of the CK 6-BAP
(6-Benzylaminopurine, 100 μ M), the competitive glucose inhibitor 2-DG (2-deoxyglucose,
2.5 mM) (**A**), or the ATP synthesis inhibitor OM (oligomycin, 1 μ M) (**B**) and incubated at $22 \pm$
2°C in the dark. Mycelia area was measured after 5 days. Floating bars are shown with
minimum maximum values, line in bar indicates median. \pm SEM, $N=10$.

**A:** 2-deoxyglucose (DG). Results were analyzed for statistical significance using a one-way
ANOVA with a Tukey post-hoc test. Lower case letters indicate statistically significant
differences between samples, with number tags indicating the group that was
comparatively analyzed, $p<0.025$. Upper case letters within the top of CK bars indicate
statistically significant differences in the level off CK-mediated growth inhibition, $p<0.018$.
Upper case letters within the bottom of DG bars indicate statistically significant differences
in the level off DG-mediated growth inhibition, $p<0.011$.

**B:** Oligomycin (OM). Results were analyzed for statistical significance using a one-way
ANOVA with a Tukey post-hoc test, or a two-tailed t-test with Welch's correction. Letters

indicate statistically significant differences between samples, with tags indicating the group
that was comparatively analyzed, $p < 0.038$.

**Fig. 6 CK promotes glucose uptake.**

*Bc* spores (10^6 / mL dissolved in sterile water) were grown in PDB (**A**) or Synthetic medium
(**B**) with 150 rpm shaking, at 22 ± 2 °C in the dark, without (Mock) or with the addition of the
CK 6-BAP (6-Benzylaminopurine, 100 μ M), or the structural control Adenine, 100 μ M. The
amount of glucose in the media was examined after 48 h, and subtracted from the amount
of glucose present in media without fungi that underwent similar treatment. The
approximate percent of increase in glucose uptake in the presence of CK is indicated above
the bars for each media concentration. Floating bars are shown with minimum maximum
values, line in bar indicates median. N=6. Results were analyzed for statistical significance
using two-tailed t-test with Welch's correction. Letters indicate statistically significant
differences between samples, **A** $p < 0.04$, **B** $p < 0.049$.

**Fig. 7 CK alters *Bc* redox state in rich media**

The redox state of *Bc* without (Mock), or in the presence of CK, was assessed using roGFP
transformed *Bc*. Spores (10^6 / mL) of *Bc* strains expressing GRX-roGFP, for assessing cytosolic
redox, and mito-roGFP, for assessing mitochondrial redox, were incubated in PDB without
(Mock) or with CK 6-BAP (6-Benzylaminopurine, 100 μ M), for 24 h at 18°C, with 150 rpm
shaking. Fluorescence was measured using a fluorimeter, with excitation at 405 ± 5 nm for
the oxidized state and 488 ± 5 nm for the reduced state of roGFP2. The emission was
detected at 510 ± 5 nm. The redox ratio of the fungus was calculated as Em_{405}/Em_{488} of
Relative fluorescence units (RFU). (**A**) Redox status of the mitochondria and cytosol, with
and without CK, after 24 h. Boxplots are shown with minimum to maximum values, inner
quartile ranges (box), median (line in box), and outer quartile ranges (whiskers), N=6. (**B**)
Time course of redox state in the cytosol. (**C**) Time course of redox state in the
mitochondria. Asterisks indicate statistical significance in a two-tailed t-test, * $p < 0.05$,
** $p < 0.01$, **** $p < 0.0001$.

**Fig. 8 Plant endogenous CK alters *Bc* redox state during infection**

The redox state of *Bc* when infecting leaves of different CK-content tomato genotypes was
assessed using roGFP transformed *Bc*. Spores (10^6 /mL in glucose and K_2HPO_4) of *Bc* strains
expressing GRX-roGFP, for assessing cytosolic redox, and mito-roGFP, for assessing
mitochondrial redox, were used to infect the background M82 wild-type line, the high-CK
*pBLS>>IPT7* overexpressing line ("IPT"), and the low-CK *pFIL>>CKX3* overexpressing line
("CKX"). *Bc* fluorescence was captured using a confocal laser scanning microscope at 24 h
and 48 h, with excitation at 405 nm for the oxidized state and 488 nm for the reduced state
of roGFP2. The emission was detected using a 505-530 nm bandpass filter. The redox ratio

of the fungus was calculated as Em405/Em488 using ImageJ, from at least 12 images per
time point, per treatment. **(A)** Redox status of the *Bc* cytosol, after 24 h and 48 h. **(B)** Redox
status of the *Bc* mitochondria, after 24 h and 48 h. **(A-B)** Floating bars are shown with
minimum to maximum values, lines indicates median, N=12. Differences between samples
were assessed using a one-way ANOVA with a Dunnett post hoc test. Different letters
indicate statistically significant differences between samples, (A) $p < 0.021$, (B) $p < 0.029$. **(C)**
Representative images of the roGFP fungi growing on leaves of the different genotypes,
captured at the "reduced" and "oxidized" wavelengths, at 48h.

**Fig. 9 Graphical model summarizing the dual role of CK in Plant-fungi interactions**

CK can signal for both growth inhibition or metabolic increases in the fungus, dependent on
nutrient availability. Under high nutritional status, the growth inhibitory effects are more
dominant, with CK causing cell cycle arrest in the fungus, resulting in less fungal growth and
therefore weaker plant infection. Under low nutrient status, the positive metabolic effects
are more dominant, leading ultimately to stronger plant infection. Model created with
BioRender.com.

We wish to thank the Editor and Reviewer's for the careful and thorough analysis of our manuscript, which has helped us improve our work. Follows a "point-by-point" response to all comments and questions included in the review.

Reviewer #1 (Comments for the Author):

In this manuscript, a wealth of experiments have been carried out on the possible dual effects of cytokinin on *Botrytis cinerea*, and the research angle is very novel, which is promising to find the key signaling substances when Bc infects plants. However, the writing of this manuscript has obvious defects:

1, Cytokinins are plant-specific chemical messengers (hormones) that play a central role in the regulation of the plant cell cycle and numerous developmental processes (T. Schmülling, in Encyclopedia of Biological Chemistry (Second Edition), 2013). This article does not specify anywhere which cytokinin is, and does not provide a molecular formula or commodity number of the cytokinin.

We used the CK 6-BAP, as indicated in the methods sections and figure legends. Additional information was added to the methods section and some of the legends, to clarify this point.

2, Regarding the experimental design based on PDA, I personally have some doubts. First, the author did not specify the preparation method or manufacturer of PDA; second, the composition of PDA is complex, and precipitation may occur after sterilization, and there are interference factors when explaining the effect of CK; third, even in the context of PDA, when the nutrient composition is reduced to 1/4 and 1/8, the bacterial growth will be severely inhibited, the growth rate and metabolic level will be significantly reduced without CK treatment, and the individual believes that there is a minimum threshold for the reduction of its physiological level, that is to say, the weakening of the inhibitory effect of CK treatment may be caused by the impending touch of the minimum threshold, and the intensity of CK active regulation is difficult to directly prove in this study.

We have added to the manuscript an extensive series of experiments conducted in synthetic media, as per the comments of the editor and all three reviewers. The advantage of synthetic media is that the exact concentration of sugars and elements is defined, and we were able to manipulate each component individually. We believe these experiments better address the question of the dependence of CK activity in *B. cinerea* on energy levels available to the fungus. In these results, we show that CK increases fungal growth when sugar is reduced to 25% of optimal levels, but when nitrogen and phosphate are reduced to 25% of optimal levels, CK inhibits growth. We added the preparation process of PDA to the methods section.

3, There are many problems in the description of methods in this manuscript, such as unknown source of strains, unknown source of reagents and instruments, and lack of corresponding references, and the specific problems have been labeled in the manuscript.

We added as much information as we could to the methods section. Notably, we did not receive, and were not able to find anywhere on the system, a marked up manuscript with comments from the reviewer.

4, The discussion section did not discuss enough relevant research within the field. Because of completely new research findings?

Indeed, there is very little information in this field. We tried to discuss all previous results as best we could.

5, There are still many missing information in this manuscript, which have been labeled in the manuscript.

We went through the manuscript and tried to add information wherever we could. Again, unfortunately, we were not able to find an upload on the system from the reviewer, so could not address any additional comments present therein.

Reviewer #2 (Comments for the Author):

Dear Authors,

First let me start with, I find the concept and the possibility that CK's influence metabolic functions in Bc depending on the nutrient (glucose) status of the environment intriguing.

Unfortunately, I have major concerns about the methods used and thus about the resulting data. If the authors can show that the observed effects are repeatable in a defined minimal medium with constant salt concentrations but variable glucose concentrations, I would be happy to reconsider my opinion.

Many thanks. In the revised version of the manuscript, we have done exactly this. We have added to the manuscript an extensive series of experiments conducted in synthetic media, as per the comments of the editor and all three reviewers. The advantage of synthetic media is that the exact concentration of sugars and elements is defined, and we were able to manipulate each component individually. We believe these experiments better address the issue of the dependence of CK activity in *B. cinerea* on energy levels available to the fungus. In these results, we show that CK increases fungal growth when sugar is reduced to 25% of optimal levels, but when nitrogen and phosphate are reduced to 25% of optimal levels, CK remains inhibitory. Experiments in defined media were conducted to assess growth (new Figure 1), growth under treatment with the competitive glucose inhibitor DG (new Figure 4), and glucose utilization (new Figure 5). The F-actin expressing fungus has abnormal growth by definition, as previously published, and does not grow well. We were not able to grow it in defined media, as it likely requires a particular element or elements that we are not aware of. Given that it would require an extensive series of experiments to generate a defined media suitable for this transgenic fungus, and that the synthetic media results confirm that the glucose availability is the key factor in determining the nature of CK influence on the fungus, we feel that the results with the F-actin expressing fungus can remain in the manuscript. We have amended the discussion concerning this aspect a bit, to clarify the message and better integrate it with the new results.

Here are the main reasons, why I can't give a positive review:

In my opinion the usage of an undefined medium such as PDB or PDA and reducing its strength by dilution leads to changes of too many variables (glucose, nitrogen, phosphate, and many more) at the same time, which makes it impossible to directly link the observed phenotypes to glucose (which is the main statement of the manuscript). The one time a defined minimal medium was used it again was diluted, thus diluting glucose amounts, but also reducing the amounts of important minerals/salts such as potassium, phosphate, sulfur and nitrogen. Thus, the data interpretation is very difficult. I'm sure lower concentrations of nitrogen and especially phosphate will also influence glycolysis or mitochondrial functions related to energy production for example.

Please repeat everything with a defined minimal medium where only the glucose concentration is changed and the salt concentrations are kept constant. Or rewrite the manuscript with a reduced focus on glucose (which would devalue the information of

the manuscript).

Many thanks. As stated above, we have repeated the key experiments in a defined medium. The reviewer will note that we had 4 types of defined media: (1) "Full" defined medium; (2) defined medium with 1/4 the amount of glucose included in the full medium; (3) defined medium with 1/4 the amount of the nitrogen and phosphate included in the full medium; and (4) defined medium where both the glucose and the nitrogen and phosphate are diluted to 1/4 the amount included in the full medium. In the context of glucose, the full defined medium (Medium 1) contained 20 g/L of glucose, which is similar to the concentration in PDB prepared according to the manufacturer's instructions. Medium 3 also contained 20 g/L glucose, while Media 2 and 4 contained 5 g/L glucose. Thus, Medium 1 is somewhat equivalent to full PDB, while Medium 4 is somewhat equivalent to 1/4 PDB, though, of course, the defined media is missing many components present in PDB.

Other points: why is sometimes boxplots with whiskers used other times only boxplots without whiskers. Are the data shown as median (not a scientific common way to show data) as state most of the times or as averages? Why are standard deviations missing for example Fig 7 B and C? Why are sometimes the individual data points included other times not?

We used the best mode to showcase the data, in our opinion, but of course, all the graphs can be changed. For boxplots, the line in the box normally represents median, this is the standard, and, in fact, cannot be changed to represent the mean in Prism, the software we used to generate the graphs. We have added a "+" sign to indicate the mean. The graphs the reviewer referred to as "boxplots without whiskers" are actually not boxplots, but rather, floating bars- meaning that the "whiskers" are included in the bars, such that the minimum to maximum values are plotted, and not quartiles. We chose floating bars in these cases because some of the samples had low values and small standard deviations, such that the boxplot becomes less informative because it is "squashed" on the X axis (boxplots do not look good with an axis broken into several segments). Displaying all points was chosen when the distribution was of interest, but the boxplots looked again less informative (when standard deviations were either very small or very large). **Given that many of the figures were replaced or changed, we tried to make the presentation as uniform as possible in the revised version of the manuscript.**

Fig 1: please redo with minimal medium where only the glucose concentration is changed.

Amended as suggested.

Fig 2: is 6h or 24h shown? Or is one timepoint missing from the graph?

The legend was corrected, thanks. 6h were required for growth in full media, and 24h for growth in minimal media, to achieve similar sizes of germlings for analysis. As stated above, the F-actin expressing fungus has abnormal growth. Given that its growth is severely stunted, any conclusion we draw takes this into account, and we are only comparing growth with and without CK under a defined set of conditions. Under these conditions, mis-localization of actin is indeed much higher in 1/4 PDB, when comparing the Mock samples. This renders the effect of CK smaller, but, there is no

difference in actin mis-localization in the two CK samples, despite the different growth times. This would suggest that the determining factor is that of the media. Actin polarization is required for growth, and when CK is applied, or when sugar/ nutrients are low, actin is less polarized, and growth is lower. We have amended the manuscript to reflect this message: that CK activity and low sugar may inhibit growth in part via a similar mechanism.

Fig 3: I find this correlation of environmental Bc strains with somewhat different growth rates and the influence of CK not very strong. There could be a multitude of reasons for those strains to behave different, which are not directly linked to CK. The authors should have used mutants in B05.10 or T4 for glucose related functions, uptake, glycolysis, TCA, ETC.. to show the impact of CK on glucose related functions (Best scenario). If this is not possible other strains with reduced growth rate, but with defined known mutations could be used as well. In Figure B the R² is missing, which would show the fitting of the curve with the data. If the one high value would be missing, the line would be pretty flat and the resulting correlation would not be significant.

We agree with the reviewer that many factors likely influence the fungal response to CK. This is not central to the story, and was moved to the supplemental data. R² is now included, and at 0.42, does suggest that the fit may be more than just a coincidence. Notably, removal of the one high point, as mentioned by the reviewer, still results in a significant correlation at p<0.03. We do think response to CK may be tied to growth rate, and have added a statement to the manuscript suggesting that additional factors likely influence the fungal response to CK as well.

Fig 4: (B) I'm confused about the p value for either the PCR or RNAseq part of the graph. Is it for the whole group of genes as an average or? Please mark the significance for each individual gene. Further for the RNAseq results what is the unit of the y-axes? Is it log or relative regular fold change? If it is relative, why are the changes not even close to be similar to the RT PCR results, where are the standard deviations? I agree the pattern is similar between the two analyses, but the fold changes are not similar. Further was the RNA DNase treated?

Many thanks. We have amended the Figure as per the reviewers suggestions, Including the p-value for each gene, the SE, and amending to actual Fold change (indeed, log(2) FC was presented in the graph previously). The RNA was DNase treated in both cases, and this was added to the methods section.

Fig 5: Is it again mean or average, no whiskers,....? The recorded data and the resulting shown pattern of 1/4 media with DG is similar to the other pattern with higher media strength, but the spread of the data especially of DG and DG+CK make any interpretation even with statistics very difficult. In my opinion not a strong indicator that DG has an influence on CK function

We have replaced this Figure with assays conducted using defined media, and moved the PDA results to the supplemental materials. Graph presentation methods were discussed above, and are now more uniform throughout the manuscript.

Fig 6: For this kind of analysis and the conclusion that CK leads to differences in glucose uptake only glucose uptake studies with labelled glucose should be used. There could be many reasons why glucose is less abundant in the CK treated condition, higher binding capacity of glucose in the fungal cell wall, by extracellular proteins or formation of a glucose containing product, which can't be measured with the reducing sugar assay,... Best to use intact protoplasts for the uptake assay with labelled glucose.

We are not currently set-up for labeled glucose assays. In the revised manuscript, we included assays with different defined media, and show that the remaining glucose in the media follows a similar trend to fungal growth in the different media, with the interesting exception of low N and P media with high glucose levels preventing CK from promoting the elimination of glucose from the media. The results in Figure 7 (previously Figure 8) also support the idea that CK alters fungal metabolism. Of course, we cannot rule out the reviewers' suggestions, and have amended the text to qualify our results and reflect these possibilities.

Fig 7: Again median? Why no standard deviations in B and C? What is the number in C at 4h. Further, especially the differences in C at 24h are so small, that they are barely visible, which questions the biological relevance.

Many thanks for this comment. This assay, now presented in Figure 6, was done *in vitro* to examine the effect of CK on fungal redox in the absence of a plant. The assays in Figure 7 (previously Figure 8) were done *in planta* with the redox-labeled fungus. Here, in Figure 6, we wanted to examine the effect of CK without the plant, and were not surprised to find that it is quite small. The time course analyses in B and C were provided to show that it required 24 h for the redox to significantly change. We have now included error bars, which are so small they are barely visible in some cases. The statistics are robust because the SDs are very small.

Fig 8: median? Whiskers?

Figure was amended as suggested.

Reviewer #3 (Comments for the Author):

The manuscript is a follow up on an earlier work in which the authors demonstrated that external application of the cytokinin 6-Benzylaminopurine (CK) can inhibit fungal growth. In the current study they used *Botrytis cinerea* to investigate the possible effect of CK on fungal metabolism and nutrition in vitro and on plants. They report that along with growth inhibition, CK promoted fungal metabolism and that these effects were inversely correlated with nutrient and sugar availability. Based on these findings, the authors propose that CK acts as "a signal to the fungus that plant tissue is present, causing it to activate energy metabolism pathways to take advantage of the available food source, while at the same time, CK is employed by the plant to inhibit the attacking pathogen". This is an intriguing concept with potential to highlight a largely ignored direction in plant microbe interactions.

While the phenomenon is interesting, there are many open questions that need to be answered. For example, how does the fungus sense CK while on the leaf? If only after plant penetration, then CK might not affect the early stages that include attachment, germination and penetration. What are the levels of CK in different types of tissues, in particular fruits and flowers, which are the most sensitive organs? Is CK evenly distributed throughout the tissues, how does (if it does?) the fungus sense it and how does it enter the fungal cell? These and similar questions warrant further research to validate the new concept. Additionally, there are some technical aspects that should be examined and it reflects on the interpretation of some results as specified below.

Many thanks for these comments. These concepts have naturally not all been addressed in our current work, though we do discuss them in the last part of the discussion, which we have amended a bit in the revised version of the manuscript to further address these issues. In general, tomato leaves can contain quite high levels of CK (100-400 nM transZeatin), which can also likely be present on the leaf surface in small amounts following micro-abrasions or minor injuries. Entry of CK into fungal cells can occur via transporters and/or endocytosis, and we have added a new Figure (Figure 8 in the revised manuscript) to attempt to examine these questions, though, we did not find that CK penetration was significantly affected by nutritional status.

Overall, this is an intriguing paper, but more work is needed to validate the new concept and part of the methods and result need careful evaluation.

Main comments

Lines 28-31: "indicating that nutrient availability is indeed the switch determining the role of CK. Transcriptomic data further support our findings, demonstrating significant upregulation to glycolysis, oxidative phosphorylation, and sucrose metabolism, upon CK treatment. Thus, the effect of CK in fungal biology depends on energy status". This statement concludes that nutrient availability and specifically energy level determine CK effects. At present the data are insufficient to draw this conclusion since they are all circumstantial, and in some cases might not support this statement. This and similar statements and conclusion throughout the manuscript should be avoided, or at least drastically toned down.

We have amended the manuscript as suggested. We believe that in its revised form, the idea that sugar levels can affect the influence of CK on the fungus is better supported than it was before.

Use of PDA: the central point of this work is possible connection between energy (carbon) levels and CK activity. PDA is an undefined medium (an extract of potatoes). It contains a wide range of metabolites other than C and N, for example amino acids, vitamins, elements etc. Each of these molecules can potentially affect CK activity in a range of ways. Therefore, the choice of PDA as the main nutrient medium is not optimal. It will be necessary to verify the phenomenon on defined media, such as CD, GH, etc., with different types of sugars.

Many thanks for this comment. In the revised version of the manuscript, we have done exactly this, as suggested. We have added an extensive series of experiments conducted in synthetic media to the manuscript, as per the comments of the editor and all three reviewers. The advantage of synthetic media is that the exact concentration of sugars and elements is defined, and we were able to manipulate each component individually. We believe these experiments better address the issue of the dependence of CK activity in *B. cinerea* on energy levels available to the fungus. In these results, we show that CK increases fungal growth when sugar is reduced to 25% of optimal levels, but when nitrogen and phosphate are reduced to 25% of optimal levels, CK remains inhibitory. Experiments in defined media were conducted to assess growth (new Figure 1), growth under treatment with the competitive glucose inhibitor DG (new Figure 4), and glucose utilization (new Figure 5).

Line 141 and similar/ Figure1: "CK mediated growth inhibition is affected by nutrient availability". I am not convinced that this statement is correct. The data in Figure 1 show that the main effect on growth is medium concentration. The way the data are presented overlooks this fact. It is hard to calculate percentage based on the graphs, but by a rough estimate it seems that when expressed as % of control, CK has similar effects (around 50% inhibition) up to 1/4 strength. At 1/8 strength the growth is already minimal and the CK effect is therefore marginal. Additionally, as already suggested, these experiments should also be conducted on defined media.

As indicated above, we have conducted these experiments in a series of defined media. The percentages of CK mediated growth inhibition or promotion were added to the Figure.

Additional comments Figure1: 1) Figure title is not necessarily supported by the data, 2) Under normal conditions *B. cinerea* grows very uniformly with minimal variance; What is the reason for the very high variance of growth measurements within treatments? 3). What you really measure is radial growth. Better present radius values, the area only inflates the numbers and might be misleading. 4) Would be good to test additional CK concentrations.

Many thanks. The revised version of this Figure deals with most of these comments. Additional CK concentrations were examined extensively in our previous work (<https://doi.org/10.1128/mBio.03068-20>), where nutrient levels were not altered. For

this work, we selected the level of CK that proved most inhibitory in the context of growth, in order to examine the full effect.

Figure 2: 1) The growth period on full and 1/4 strength media differs considerably; this factor can have immense impact on results, possibly far greater than the effect of CK. 2) At 1/4 strength the wild type also shows a lot of un-oriented actin namely there is a strong effect of the medium. Additionally, the way actin is counted is not necessarily reflecting changes due to CK effect. 3) What about results in 1/2 PDB? 4) What was CK concentration?

The F-actin expressing fungus has abnormal growth by definition, as previously published, and does not grow well. We were not able to grow it in defined media, as it likely requires a particular element or elements that we are not aware of. Given that it would require an extensive series of experiments to generate a defined media suitable for this transgenic fungus, and that the synthetic media results confirm that the glucose availability is the key factor affecting the nature of CK influence on the fungus, we feel that the results with the F-actin expressing fungus grown in PDB is sufficiently informative to remain in the manuscript. We reasoned that comparing actin distribution in mycelia of similar sizes was better than growing them for the same amount of time, because if the mycelia did not grow at all in 1/4 PDB after 6h, the comparison of the actin localization is not valid. As growth of the actin expressing fungus is severely stunted, any conclusion we draw takes this into account, and we were chiefly interested in comparing growth with and without CK. Mis-localization of actin is indeed much higher in 1/4 PDB. This renders the effect of CK smaller, but, there is no difference in actin mis-localization in the two CK samples, despite the different growth times. This indeed indicates that the determining factor is that of the media strength. Actin polarization is required for growth, and when CK is applied, or when sugar is low, actin is less polarized, and growth is lower. We have amended the manuscript to reflect this message: that CK activity and low sugar may inhibit growth in part via a similar mechanism.

Figure 3: 1) Correlation is positive, but weak and probably doesn't mean much.

This Figure was moved to the supplemental materials. It is likely that many factors can affect the growth rate and response to CK of different *B. cinerea* isolates. We do think response to CK may be tied to growth rate, and have amended this part of the manuscript to include additional possibilities.

Minor comments

Line 22: When referring to the scientific name of an organism it is customary to write the first letter of the genus and the full name of the species. Suggest replacing all Bc with *B. cinerea*

Amended as suggested.

Lines 68-69: Delete: "CK was also found to enhance disease resistance to additional, non obligatory plant pathogens". It is repeated in the following sentence.

Amended as suggested.

Lines 70-71: What do you mean by "endogenous application"?

Corrected, thanks.

Line 130-131: This is probably one of the strongest result, but it does not necessarily relate to the strength of the medium.

This part of the manuscript was amended - see also above.

Lines 189-190: This might not be the correct interpretation: the fact that there was no further inhibition doesn't necessarily mean the fungus was insensitive to CK.

This part of the manuscript was amended.

Line 417: What is the source of this medium? What is the N source?

The different media used in this work are now detailed in Table 1 in the methods section.

Line 446: If you did not carry additional RNAseq experiment, please explicitly mention that analyses were performed on old data.

Added, thanks.

Figure 5: 1) Reasons for high variance are unclear. 2) The strong effect of OM might mask the CK effect. Would be good to test at lower OM concentrations.

This Figure was replaced, with the PDA results moved to the supplemental material. The variance is not high in the Mock samples, or in rich media, but does seem to increase when stressors (CK or glucose inhibitors) are applied. This is very interesting and could be investigated in the future. We tested several OM concentrations prior to conducting the experiment, and chose the one that allowed fungal growth but still inhibited it (25-50%). We have added percentage levels to the Figure, which is now in the supplement.

Figure 6: 1) Which medium was used and what was the main carbon source? 2) Presentation of % (top) might be misleading since the actual values are very low.

This Figure was replaced, and the PDA results moved to the supplemental material.

May 23, 2022

Dr. Maya Bar
Agricultural Research Organization
Plant Pathology and Weed Research
Bet Dagan
Israel

Re: Spectrum00280-22R1-A (Cytokinin regulates energy utilization in *Botrytis cinerea*)

Dear Dr. Maya Bar:

Two experts that reviewed the previously submitted version of your manuscript find the manuscript improved. Reviewer 2 raised 2 main points on pH and growth condition, please respond to the queries raised.

Link Not Available

Sincerely,

Giuseppe Ianiri

Journals Department
Reviewer comments:

Reviewer #1 (Comments for the Author):

This article is well written now, but I still suggest that it is more appropriate to replace CK with specific 6-BAP.

Reviewer #2 (Comments for the Author):

Dear Authors,

I'm happy the results could be repeated with a defined medium. Thus, it makes it far easier to judge the manuscript. Further,

thank you for already fixing the issues and suggested changes from the earlier unrevised version.

I still have a few questions about the media and growth conditions.

What pH was used for the minimal medium? PDB usually has a pH of 5.1-5.4 was it the same for the minimal medium?

Further, as the growth in the minimal medium between the different treatments was different, was the pH of the supernatant at the endpoint of the treatments checked for the different treatments and was it similar? Fungi usually acidify the media under glycolysis and acidification could change the activity of CK and this could be dependent on the growth. If no difference of the pH was overserved for the different treatments, that would be fine. If difference were observed, maybe a quick analysis of the impact of pH on the CK effect should be conducted.

Another observation I made was, that all the experiments were performed in the dark. This is ok, as it would simulate the night. But I wonder, as virulence of Botrytis is light dependent, if nutrient acquisition and usage is light dependent too? Thus, could the authors repeat the growth assay with Ck in the light for the 3 days? If the results are similar, no impact of light, that would be fine. If not, that could change some of the statements of the paper. Also, a night/ day scenario could be analyzed.

Minor issues:

Maybe change "salt" and "nutrient elements" to minerals in the manuscript. This should give the reader a better distinction between sugar and macronutrients such as N and PO₄.

I think there should be a "state" included in the main text and figure legends every time cytosolic redox or mitochondrial redox (state) is used?

Line 38: "several works", not sure these words have the correct meaning

In the reference 39 and 40 are identical

Figure 3 botrytis please capitalize

Staff Comments:

Preparing Revision Guidelines

Please return the manuscript within 60 days; if you cannot complete the modification within this time period, please contact me. If you do not wish to modify the manuscript and prefer to submit it to another journal, please notify me of your decision immediately so that the manuscript may be formally withdrawn from consideration by Microbiology Spectrum.

We wish to thank the reviewers for their continued attention and time invested in our manuscript. Follows a "point by point" response to the additional issues raised.

Reviewer #1 (Comments for the Author):

This article is well written now, but I still suggest that it is more appropriate to replace CK with specific 6-BAP.

We do not disagree with the reviewer on this point, of course their suggestion is more accurate. However, we feel that changing the easily recognizable "CK" to the molecule name will make it confusing for readers outside our field. We have previously published results showing that different CK molecules have similar effects on *B. cinerea* (DOI: <https://doi.org/10.1128/mBio.03068-20>). We have now added the chemical name to the manuscript at key points, and it appears in all the Figure legends.

Reviewer #2 (Comments for the Author):

Dear Authors,

I'm happy the results could be repeated with a defined medium. Thus, it makes it far easier to judge the manuscript. Further, thank you for already fixing the issues and suggested changes from the earlier unrevised version.

I still have a few questions about the media and growth conditions. What pH was used for the minimal medium? PDB usually has a pH of 5.1-5.4 was it the same for the minimal medium?

The synthetic media had a pH of about 6.5. We avoided trying to alter the pH, since glucose and minerals in water are a very poor buffer. Our PDB is usually around 5.5, making this synthetic media more basic, however, we did verify that the fungus was able to grow in these media before performing the assays.

Further, as the growth in the minimal medium between the different treatments was different, was the pH of the supernatant at the endpoint of the treatments checked for the different treatments and was it similar? Fungi usually acidify the media under glycolysis and acidification could change the activity of CK and this could be dependent on the growth. If no difference of the pH was overserved for the different treatments, that would be fine. If difference were observed, maybe a quick analysis of the impact of pH on the CK effect should be conducted.

We now include new supplemental Figure S4. As suggested by the Reviewer, we examined changes to medium pH in all our assays, to confirm that the observed growth effects are not a result of pH alterations. CK promoted media acidification in all cases. Although media acidification was slightly stronger in media #3 and #4 as compared with media #1 and #2, levels were similar in full media (media #1) and media with restricted glucose (media #2), suggesting that the media pH is not the source of the observed alterations in CK activity upon glucose restriction. As with our glycolysis results, we do see that diluting the minerals can affect media acidification, however, this is a different story not necessarily related to the central glucose story of the

manuscript, than requires extensive further investigation. While we intend to undertake this in the future, we see it as a separate story.

Another observation I made was, that all the experiments were performed in the dark. This is ok, as it would simulate the night. But I wonder, as virulence of Botrytis is light dependent, if nutrient acquisition and usage is light dependent too? Thus, could the authors repeat the growth assay with Ck in the light for the 3 days? If the results are similar, no impact of light, that would be fine. If not, that could change some of the statements of the paper. Also, a night/ day scenario could be analyzed.

Many thanks for this comment. As suggested, we conducted identical assays with fungi grown in light, with very similar results, now included in new Figure S3.

Minor issues:

Maybe change "salt" and "nutrient elements" to minerals in the manuscript. This should give the reader a better distinction between sugar and macronutrients such as N and PO₄.

Many thanks for this comment, amended as suggested.

I think there should be a "state" included in the main text and figure legends every time cytosolic redox or mitochondrial redox (state) is used?

Amended, thanks.

Line 38: "several works", not sure these words have the correct meaning

Amended, thanks.

In the reference 39 and 40 are identical

Amended, thanks.

Figure 3 botrytis please capitalize

Amended, thanks.

July 11, 2022

Dr. Maya Bar
Agricultural Research Organization
Plant Pathology and Weed Research
Bet Dagan
Israel

Re: Spectrum00280-22R2 (Cytokinin regulates energy utilization in *Botrytis cinerea*)

Dear Dr. Maya Bar:

Your manuscript has been accepted, and I am forwarding it to the ASM Journals Department for publication. You will be notified when your proofs are ready to be viewed.

Sincerely,

Giuseppe Ianiri
Editor, Microbiology Spectrum

Journals Department
Supplemental Dataset: Accept
Supplemental Material: Accept